# Assessing the simulated soil hydrothermal regime of active layer from Noah-MP LSM v1.1 in the permafrost regions of the Qinghai-Tibet Plateau

Xiangfei Li[1,2,3], Tonghua Wu[1,3,*], Xiaodong Wu[1], Jie Chen[1], Xiaofan Zhu[1], Guojie Hu[1], Ren Li[1], Yongping Qiao[1], Cheng Yang[1,3], Junming Hao[1,3], Jie Ni[1,3], Wensi Ma[1,3]

[1] Cryosphere Research Station on the Qinghai-Tibet Plateau, State Key Laboratory of Cryospheric Science, Northwest Institute of Eco-Environment and Resources, Chinese Academy of Sciences, Lanzhou 730000, China

[2] National Cryosphere Desert Data Center, Northwest Institute of Eco-Environment and Resources, Chinese Academy of Sciences, Lanzhou 730000, China

[3] University of Chinese Academy of Sciences, Beijing 100049, China

**Correspondence:** Tonghua Wu (thuawu@lzb.ac.cn)

**Abstract.** Extensive and rigorous model inter-comparison is of great importance before application due to the uncertainties in current land surface models (LSMs). Without considering the uncertainties of forcing data and model parameters, this study designed an ensemble of 55296 experiments to evaluate the Noah land surface model with multi-parameterization (Noah-MP) for snow cover events (SCEs), soil temperature (ST) and soil liquid water (SLW) simulation, and investigated the sensitivity of parameterization schemes at a typical permafrost site on the Qinghai-Tibet Plateau. The results showed that Noah-MP systematically overestimates snow cover, which could be greatly resolved when adopting the sublimation from wind and semi-implicit snow/soil temperature time scheme. As a result of the overestimated snow, Noah-MP generally underestimates ST and ST is mostly influenced by the snow process. Systematic cold bias and large uncertainties of soil temperature remains after eliminating the effects of snow, particularly at the deep layers and during the cold season. The combination of roughness length for heat and under-canopy aerodynamic resistance contributes to resolve the cold bias of soil temperature. In addition, Noah-MP generally underestimates top SLW. The RUN process dominates the SLW simulation in comparison of the very limited impacts of all other physical processes. The analysis of the model structural uncertainties and characteristics of each scheme would be constructive to a better understanding of the land surface processes in the permafrost regions of the QTP and further model improvements towards soil hydrothermal regime modeling using the LSMs.

## 1 Introduction

The Qinghai-Tibet Plateau (QTP) is underlain by the world's largest high-altitude permafrost covering a contemporary area of $1.06 \times 10^6$ km$^2$ (Zou et al., 2017). Under the background of climate warming and intensifying human activities, soil hydrothermal dynamics in the permafrost regions on the QTP has been widely suffering from soil warming (Wang et al., 2021), soil wetting (Zhao et al., 2019), and changes in soil freeze-thaw cycle (Luo et al., 2020).Such changes have not only induced the reduction of permafrost extent, disappearing of permafrost patches and thickening of active layer (Chen et al., 2020), but also resulted in alterations in hydrological cycles (Zhao et al., 2019; Woo, 2012), changes of ecosystem (Fountain et al., 2012; Yi et al., 2011) and damages to infrastructures (Hjort et al., 2018). Therefore, it is very important to monitor and simulate the soil hydrothermal regime to adapt to the changes taking place.

A number of monitoring sites have been established in the permafrost regions of the QTP (Cao et al., 2019). However, it is inadequate to construct the soil hydrothermal state by considering the spatial variability of the ground thermal regime and an uneven distribution of these observations. In contrast, numerical models are competent alternatives. In recent years, land surface models (LSMs), which describe the exchanges of heat, water, and momentum between the land and atmosphere (Maheu et al., 2018), have received significant improvements in the representation of permafrost and frozen ground processes (Koven et al., 2013; Nicolsky et al., 2007; Melton et al., 2019). LSMs are capable of simulating the transient change of subsurface hydrothermal processes (e.g. soil temperature and moisture) with soil heat conduction (-diffusion) and water movement equations (Daniel et al., 2008). Moreover, they could be integrated with the numerical weather prediction system like WRF (Weather Research and Forecasting), making them as effective tools for comprehensive interactions between climate and permafrost (Nicolsky et al., 2007).

Some LSMs have been evaluated and applied in the permafrost regions of the QTP. Guo and Wang (2013) investigated near-surface permafrost and seasonally frozen

ground states as well as their changes using the Community Land Model, version 4
(CLM4). Hu et al. (2015) applied the coupled heat and mass transfer model to identify
the hydrothermal characteristics of the permafrost active layer in the Qinghai-Tibet
Plateau. Using an augmented Noah LSM, Wu et al. (2018) modeled the extent of
permafrost, active layer thickness, mean annual ground temperature, depth of zero
annual amplitude and ground ice content on the QTP in 2010s. Despite those
achievements based on different models, LSMs are in many aspects insufficient in
permafrost regions. For one thing, large uncertainties still exist in the state-of-the-art
LSMs when simulating the soil hydrothermal regime on the QTP (Chen et al., 2019).
For instance, 19 LSMs in CMIP5 overestimate snow depth over the QTP (Wei and Dong,
2015), which could result in the variations of the soil hydrothermal regime in the aspects
of magnitude and vector (cooling or warming) (Zhang, 2005). Moreover, most of the
existing LSMs are not originally developed for permafrost regions. Many of their soil
processes are designed for shallow soil layers (Westermann et al., 2016), but permafrost
would occur in the deep soil. And the soil column is often considered homogeneous,
which cannot represent the stratified soil common on the QTP (Yang et al., 2005). Given
the numerous LSMs and possible deficiencies, it is necessary to assess the
parameterization schemes for permafrost modeling on the QTP, which is helpful to
identify the influential sub-processes, enhance our understanding of model behavior,
and guide the improvement of model physics (Zhang et al., 2016).
Noah land surface model with multi-parameterization (Noah-MP) provides a
unified framework in which a given physical process can be interpreted using multiple
optional parameterization schemes (Niu et al., 2011). Due to the simplicity in selecting
alternative schemes within one modeling framework, it has been attracting increasing
attention in inter-comparison work among multiple parameterizations at point and
watershed scales (Hong et al., 2014; Zheng et al., 2017; Gan et al., 2019; Zheng et al.,
2019; Chang et al., 2020; You et al., 2020a). For example, Gan et al. (2019) carried out
an ensemble of 288 simulations from multi-parameterization schemes of six physical
processes, assessed the uncertainties of parameterizations in Noah-MP, and further

revealed the best-performing schemes for latent heat, sensible heat and terrestrial water storage simulation over ten watersheds in China. You et al. (2020b) assessed the performance of Noah-MP in simulating snow process at eight sites over distinct snow climates and identified the shared and specific sensitive parameterizations at all sites, finding that sensitive parameterizations contribute most of the uncertainties in the multi-parameterization ensemble simulations. Nevertheless, there is little research on the inter-comparison of soil hydrothermal processes in the permafrost regions. In this study, an ensemble experiment of totally 55296 scheme combinations was conducted at a typical permafrost monitoring site on the QTP. The simulated snow cover events (SCEs), soil temperature (ST) and soil liquid water (SLW) of Noah-MP model was assessed and the sensitivities of parameterization schemes at different depths were further investigated. This study could be expected to present a reference for soil hydrothermal simulation in the permafrost regions on the QTP.

This article is structured as follows: Section 2 introduces the study site, atmospheric forcing data, design of ensemble simulation experiments, and sensitivity analysis methods. Section 3 describes the ensemble simulation results of SCEs, ST and SLW, explores the sensitivity and interactions of parameterization schemes. Section 4 discusses the schemes in each physical process. Section 5 concludes the main findings.

## 2   Methods and materials

### 2.1 Site description and observation datasets

Tanggula observation station (TGL) lies in the continuous permafrost regions of Tanggula Mountain, central QTP (33.07° N, 91.93° E, Alt.: 5,100 m a.s.l; Fig. 1). This site a typical permafrost site on the plateau with sub-frigid and semiarid climate (Li et al., 2019), filmy and discontinuous snow cover (Che et al., 2019), sparse grassland (Yao et al., 2011), coarse soil (Wu and Nan, 2016; He et al., 2019), and thick active layer (Luo et al., 2016), which are common features in the permafrost regions of the plateau. According to the observations from 2010–2011, the annual mean air temperature of

TGL site was −4.4 °C. The annual precipitation was 375 mm, and of which 80 % is concentrated between May and September. Alpine steppe with low height is the main land surface, whose coverage range is about 40 % ~ 50 % (Yao et al., 2011). The active layer thickness is about 3.15 m (Hu et al., 2017).

The atmospheric forcing data, including wind speed/direction, air temperature/relative humidity/pressure, downward shortwave/longwave radiation, and precipitation, were used to drive the model. These variables above were measured at a height of 2 m and covered the period from August 10, 2010 to August 10, 2012 (Beijing time) with a temporal resolution of 1 hour. Daily soil temperature and liquid moisture at depths of 5 cm, 25 cm, 70 cm, 140 cm, 220 cm and 300 cm from August 10, 2010 to August 9, 2011 (Beijing time) were utilized to validate the simulation results.

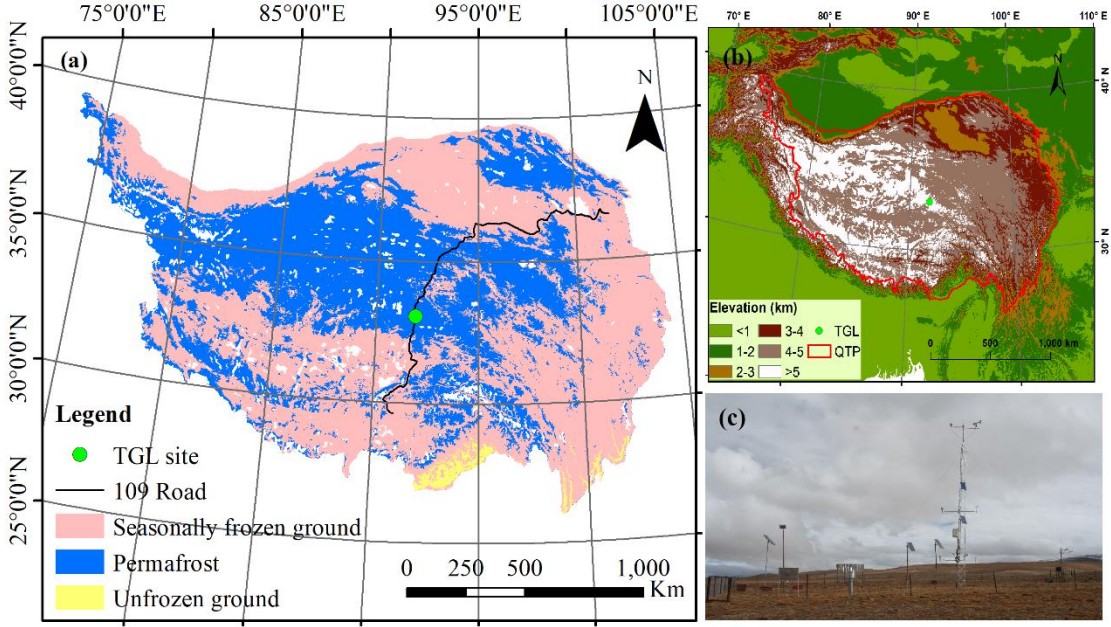

**Figure 1.** Location and geographic features of study site. (a) Location of observation site and permafrost distribution (Zou et al., 2017). (b) Topography of the Qinghai-Tibet Plateau. (c) Photo of the Tanggula observation station.

**2.2 Ensemble experiments of Noah-MP**

The offline Noah-MP LSM v1.1 was assessed in this study. The default Noah-MP consists of 12 physical processes that are interpreted by multiple optional

parameterization schemes. These sub-processes include vegetation model (VEG), canopy stomatal resistance (CRS), soil moisture factor for stomatal resistance (BTR), runoff and groundwater (RUN), surface layer drag coefficient (SFC), super-cooled liquid water (FRZ), frozen soil permeability (INF), canopy gap for radiation transfer (RAD), snow surface albedo (ALB), precipitation partition (SNF), lower boundary of soil temperature (TBOT) and snow/soil temperature time scheme (STC) (Table 1). Details about the processes and optional parameterizations can be found in Yang et al. (2011a).

VEG(1) is adopted in the VEG process, in which the vegetation fraction is prescribed according to the NESDIS/NOAA 0.144 degree monthly 5-year climatology green vegetation fraction (https://www.emc.ncep.noaa.gov/mmb/gcip.html), and the monthly leaf area index (LAI) was derived from the Advanced Very High-Resolution Radiometer (AVHRR) (https://www.ncei.noaa.gov/data/, Claverie et al., 2016). Previous studies has confirmed that Noah-MP seriously overestimate the snow events and underestimate soil temperature and moisture on the QTP (Jiang et al., 2020; Li et al., 2020; Wang et al., 2020), which can be greatly resolved by considering the sublimation from wind (Gordon scheme) and a combination of roughness length for heat and under-canopy aerodynamic resistance (Y08-UCT) (Zeng et al., 2005; Yang et al., 2008; Li et al., 2020). For a more comprehensive assessment, we added two physical processes based on the default Noah-MP model, i.e. the snow sublimation from wind (SUB) and the combination scheme process (CMB) (Table 1). In the two processes, users can choose to turn on the Gordon and Y08-UCT scheme (described in the study of Li et al., 2020) or not. As a result, in total 55296 combinations are possible for the 13 processes and orthogonal experiments were carried out to evaluate their performance in soil hydrothermal dynamics.

The Noah-MP model was modified to consider the vertical heterogeneity in the soil profile by setting the corresponding soil parameters for each layer. The soil hydraulic parameters, including the porosity, saturated hydraulic conductivity, hydraulic potential, the Clapp-Hornberger parameter b, field capacity, wilt point, and

saturated soil water diffusivity, were determined using the pedotransfer functions
proposed by Hillel (1980), Cosby et al. (1984), and Wetzel and Chang (1987)
(Equations S1-S7), in which the sand and clay percentages were based on Hu et al.,
(2017) (Table S1). In addition, the simulation depth was extended to 8.0 m to cover the
active layer thickness of the QTP. The soil column was discretized into 20 layers, whose
depths follow the default scheme in CLM 5.0 (Table S1, Lawrence et al., 2018). Due to
the inexact match between observed and simulated depths, the simulations at 4 cm, 26
cm, 80 cm, 136 cm, 208 cm and 299 cm were compared with the observations at 5 cm,
25 cm, 70 cm, 140 cm, 220 cm and 300 cm, respectively. A 30-year spin-up was
conducted in every simulation to reach equilibrium soil states.
**Table 1.** The physical processes and options of Noah-MP.

| Physical processes | Options |
|---|---|
| Vegetation model (VEG) | (1) table LAI, prescribed vegetation fraction |
| | (2) dynamic vegetation |
| | (3) table LAI, calculated vegetation fraction |
| | (4) table LAI, prescribed max vegetation fraction |
| Canopy stomatal resistance (CRS) | (1) Jarvis |
| | (2) Ball-Berry |
| Soil moisture factor for stomatal resistance (BTR) | (1) Noah |
| | (2) CLM |
| | (3) SSiB |
| Runoff and groundwater (RUN) | (1) SIMGM with groundwater |
| | (2) SIMTOP with equilibrium water table |
| | (3) Noah (free drainage) |
| | (4) BATS (free drainage) |
| Surface layer drag coefficient (SFC) | (1) Monin-Obukhov (M-O) |
| | (2) Chen97 |
| Super-cooled liquid water (FRZ) | (1) generalized freezing-point depression |
| | (2) Variant freezing-point depression |
| Frozen soil permeability (INF) | (1) Defined by soil moisture, more permeable |
| | (2) Defined by liquid water, less permeable |
| Canopy gap for radiation transfer (RAD) | (1) Gap=F(3D structure, solar zenith angle) |
| | (2) Gap=zero |
| | (3) Gap=1-vegetated fraction |
| Snow surface albedo (ALB) | (1) BATS |
| | (2) CLASS |
| Precipitation partition (SNF) | (1) Jordan91 |
| | (2) BATS: $T_{sfc} < T_{frz}+2.2K$ |
| | (3) $T_{sfc} < T_{frz}$ |

| | |
|---|---|
| Lower boundary of soil temperature (TBOT) | (1) zero heat flux (2) soil temperature at 8m depth |
| Snow/soil temperature time scheme (STC) | (1) semi-implicit (2) full implicit |
| Snow sublimation from wind (SUB) | (1) No (2) Yes |
| Combination scheme by Li et al.(2020) (CMB) | (1) No (2) Yes |

BATS (Biosphere–Atmosphere Transfer Model); CLASS (Canadian Land Surface Scheme);
SIMGM (Simple topography-based runoff and Groundwater Model); SIMTOP (Simple
Topography-based hydrological model); SSiB (Simplified Simple Biosphere model).
**2.3 Methods for sensitivity analysis**
The simulated snow cover events (SCEs) was quantitatively evaluated using the
overall accuracy index (OA) (Toure et al., 2016):
$$OA = \frac{a + d}{a + b + c + d}$$
where $a$ is the positive hits, $b$ represents the false alarm, $c$ is the misses, and $d$
represents the negtive hits. The value of OA range from 0 to 1. A higher OA signifies
better performance. Ground albedo was used as an indicator for snow events due to a
lack of snow depth observations. The days when the daily mean albedo is greater than
the observed mean value of the warm and cold season (0.25 and 0.30, respectively) are
identified as snow cover.
The root mean square error (RMSE) between the simulations and observations
were adopted to evaluate the performance of Noah-MP in simulating soil hydrothermal
dynamics.
To investigate the influence degrees of each physical process on SCEs, ST and
SLW, we firstly calculated the mean OA (for SCE) and mean RMSE (for ST and SLW)
($\bar{Y}_j^i$) of the $j$th parameterization schemes ($j$ = 1, 2, …) in the $i$th process ($i$ = 1, 2, …).
Then, the maximum difference of $\bar{Y}_j^i$ ($\Delta\overline{OA}$ $or$ $\Delta\overline{RMSE}$) was defined to quantify the
sensitivity of the $i$th process ($i$ = 1, 2, …) (Li et al., 2015):
$$\Delta\overline{OA} \ or \ \Delta\overline{RMSE} = \bar{Y}_{max}^i - \bar{Y}_{min}^i$$
where $\bar{Y}_{max}^i$ and $\bar{Y}_{min}^i$ are the largest and the smallest $\bar{Y}_j^i$ in the $i$th process,
respectively. For a given physical process, a high $\Delta\overline{OA}$ $or$ $\Delta\overline{RMSE}$ signifies large
difference between parameterizations, indicating high sensitiveness of the $i$th process
for SCEs and ST/SLW simulation.
The sensitivities of physical processes were determined by quantifying the
statistical distinction level of performance between parameterization schemes. The
Independent-sample T-test (2-tailed) was adopted to identify whether the distinction
level between two schemes is significant, and that between three or more schemes was
tested using the Tukey's test. Tukey's test has been widely used for its simple
computation and statistical features (Benjamini, 2010). The detailed descriptions about
this method can be found in Zhang et al. (2016), Gan et al. (2019), and You et al. (2020a).
A process can be considered sensitive when the schemes show significant difference.
Moreover, schemes with large mean OA and small mean RMSE were considered
favorable for SCEs and ST/SLW simulation, respectively. We distinguished the
differences of the parameterization schemes at 95 % confidence level.
**3   Results**
**3.1 General performance of the ensemble simulation**
The performance of Noah-MP for snow simulation was firstly tested by conducting
an ensemble of 55296 experiments. Due to a lack of snow depth measurements, ground
albedo was used as an indicator for snow cover. Figure 2 shows the monthly variations
of observed ground albedo and the simulations produced by the ensemble simulations.
The ground albedo was extremely overestimated with large uncertainties when
considering the snow options in Noah-MP, indicating the overestimation of snow depth
and duration. Such overestimation continued till July.

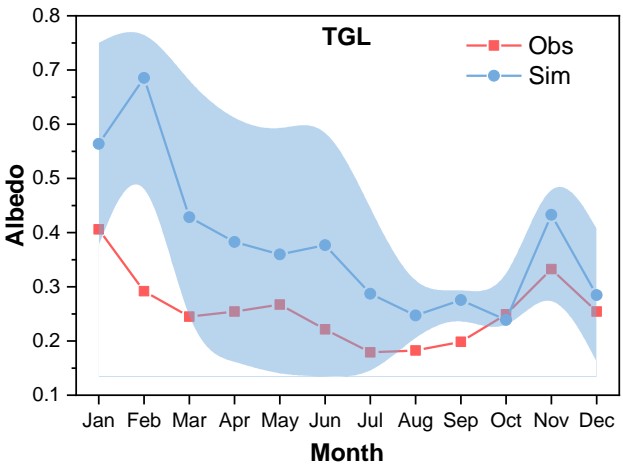

**Figure 2.** Monthly variations of ground albedo at TGL site for observation (Obs), and the ensemble simulation (Sim). The light blue shadow represents the standard deviation of the ensemble simulation.

Figure 3 illustrates the ensemble simulated and observed annual cycle of ST and SLW at TGL site. The ensemble experiments basically captured the seasonal variability of ST, whose magnitude decreased with soil depth. In addition, the simulated ST in the snow-affected season (October-July) showed relatively wide uncertainty ranges, particularly at the shallow layers. This indicates that the selected schemes perform much differently for snow simulation, resulting in large uncertainties of shallow STs. The simulated ST were generally smaller than the observations with relatively large gaps during the snow-affected season. It indicates that the Noah-MP model generally underestimates the ST, especially during the snow-affected months.

Since the observation equipment can only record the liquid water, soil liquid water (SLW) was evaluated against simulations from the ensemble experiments (Fig. 3). The Noah-MP model generally underestimated surface (5 cm and 25 cm) and deep (220 cm and 300 cm) SLW (Fig. 3g, 3h, 3k, 3l). However, Noah-MP tended to overestimate the SLW at the middle layers of 70 cm and 140 cm. Moreover, the simulated SLW exhibited relatively wide uncertainty ranges, particularly during the warm season (Fig. 3).

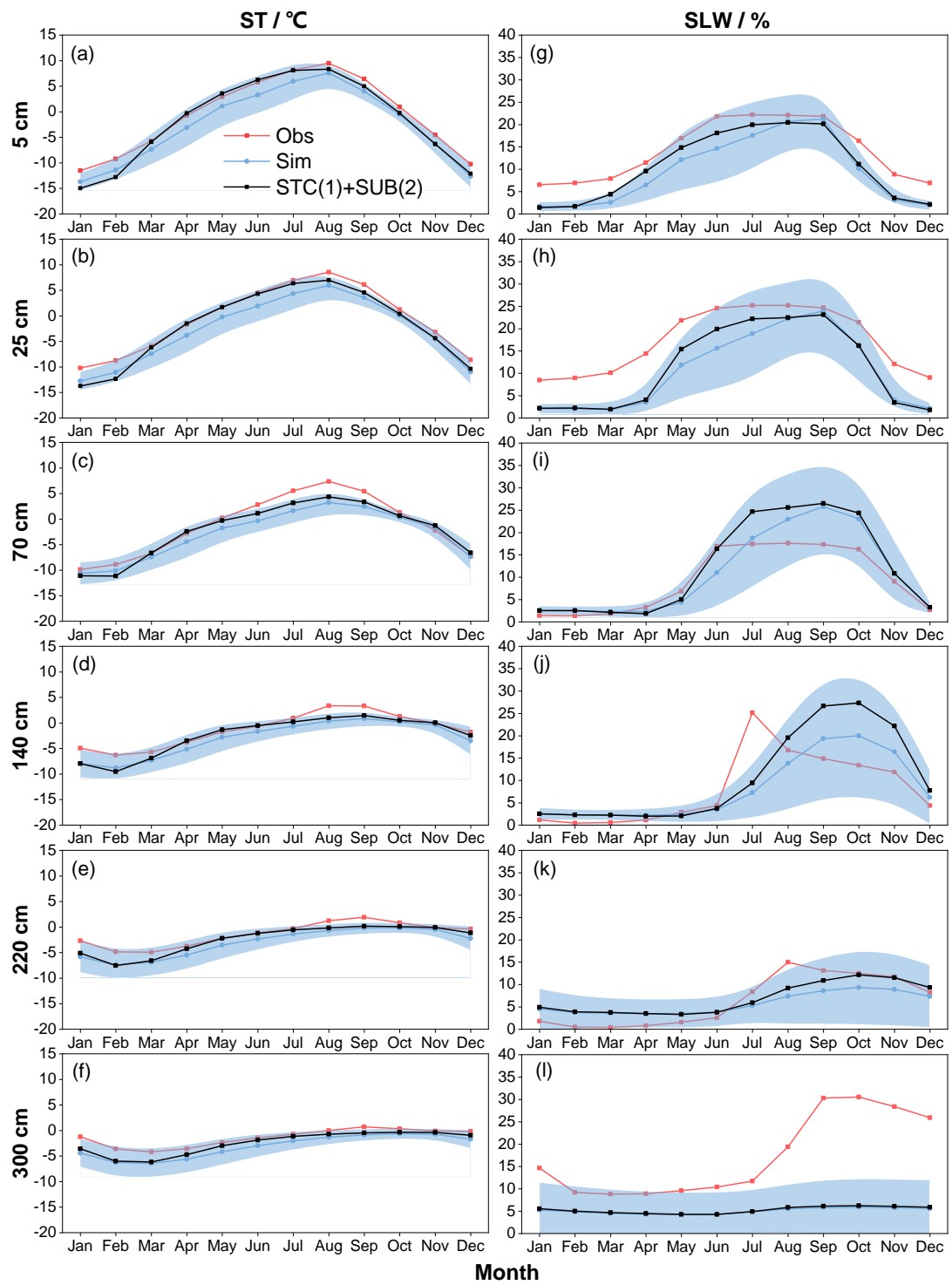

249

**Figure 3.** Monthly soil temperature (ST in °C) and soil liquid water (SLW in %) at (a,

g) 5 cm, (b, h) 25 cm, (c, i) 70 cm, (d, j) 140 cm, (e, k) 220 cm, (f, l) 300 cm at TGL

site. The light blue shadow represents the standard deviation of the ensemble simulation.

The black line-symbol represents the ensemble mean of simulations with STC(1) and

SUB(2).

**3.2 Sensitivity of physical processes**

**3.2.1 Influence degrees of physical processes**

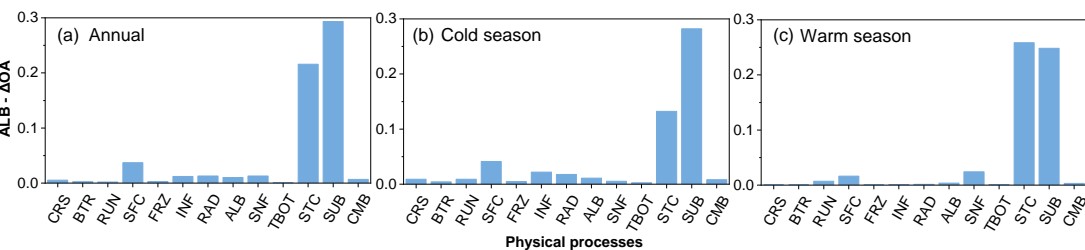

**Figure 4.** The maximum difference of the mean overall accuracy (OA) for albedo (ALB-$\Delta OA$) in each physical process during the (a) annual, (b) cold season, and (c) warm season at TGL site.

Figure. 4 compares the influence scores of the 13 physical processes based on the maximum difference of the mean OA over 55296 experiments using the same scheme, for SCEs at TGL site. On the whole, the SUB and STC processes had the largest scores for the whole year as well as during both the warm and cold seasons, and the other processes showed a value less than 0.05 (Fig. 4a, 4b, 4c). Moreover, the SUB process had a consistent influence on SCEs while the influence of STC differed with season. In the cold season, the score of SUB process (0.28) was two times more than that of the STC process (Fig. 4b), indicating the relative importance of snow sublimation for SCEs simulation during the cold season. When it comes to the warm season, the influence score of SUB (0.25) did not change much, while that of STC increased to 0.26 and showed a similar influence on SCEs simulation with SUB.

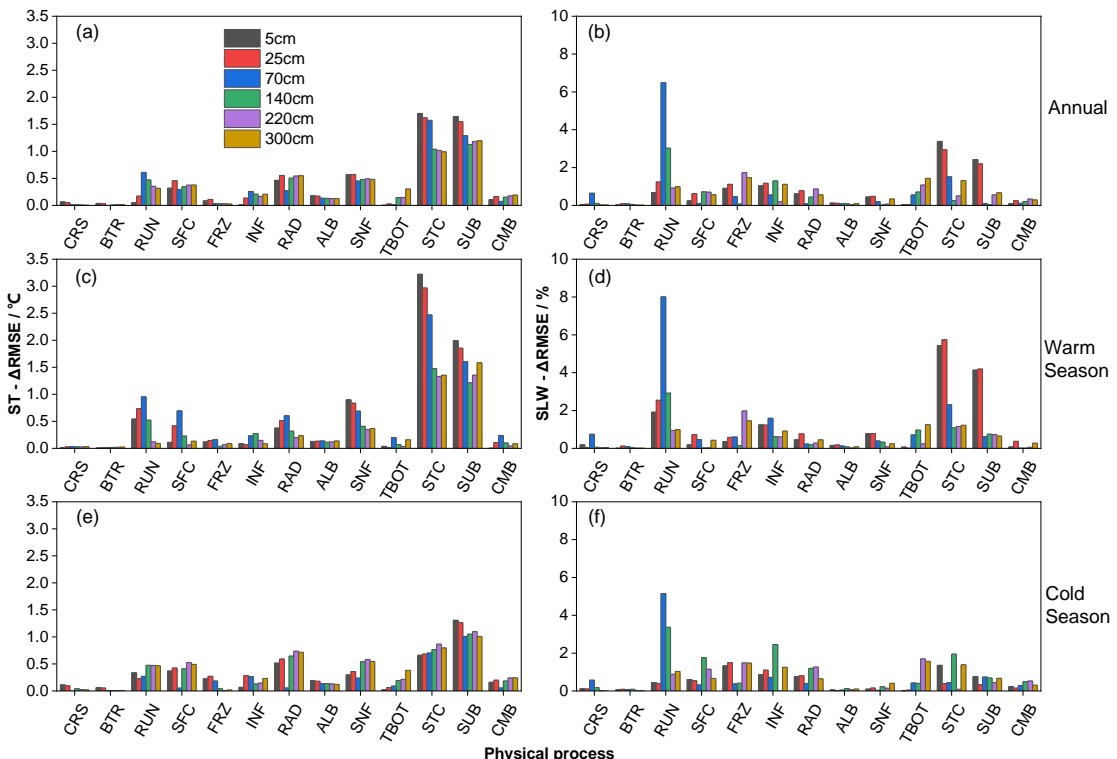

**Figure 5.** The maximum difference of the mean RMSE for (a, c and e) soil temperature (ST-$\Delta\overline{RMSE}$ in °C) and (b, d and f) soil liquid water (SLW-$\Delta\overline{RMSE}$ in %) in each physical process during the (a and b) annual, (c and d) warm, and (e and f) cold season at different soil depths at TGL site.

Figure. 5 compares the influence scores of the 13 physical processes at different soil depths, based on the maximum difference of the mean RMSE over 55296 experiments using the same scheme, for ST and SLW at TGL site. The snow-related processes, including the STC, SUB and SNF process showed the largest ST-$\Delta\overline{RMSE}$ at all layers, followed by the RAD, SFC and RUN processes. While the ST-$\Delta\overline{RMSE}$ of the other 7 physical processes were less than 0.5°C, among which the influence of CRS and BTR processes were negligible. What's more, the FRZ, INF, and TBOT processes had larger influence scores during the cold season than warm season, and the scores of TBOT were greater in deep soils than shallow soils. During the warm season, the physical processes generally showed more influence on shallow soil temperatures. When it comes to the cold season, the influence of the physical processes on deep layers obviously increased and comparable with that on shallow layers, implying the relatively higher uncertainties of Noah-MP during the cold season.

Most of the $\Delta\overline{RMSE}$ for SLW are less than 5 %, indicating that all the physical
processes have limited influence on the SLW, among which CRS, BTR, ALB, SNF, and
CMB showed the smallest effects on SLW (Fig. 5b, 5d, 5f). During the warm season,
the RUN process, together with the STC and SUB processes, dominated the
performance of SLW simulation, especially at shallow layers (5 cm, 25 cm and 70 cm,
Fig. 5d). During the cold season, however, the RUN process dominated the SLW
simulation with a great decline of dominance of STC and SUB processes.

**3.2.2 Sensitivities of physical processes and general behaviors of**

**parameterizations**

To further investigate the sensitivity of each process and the general performance
of the parameterizations, the Independent-sample T-test (2-tailed) and Tukey's test were
conducted to test whether the difference between parameterizations within a physical
process is significant (Fig. 6 and 7). In a given sub-process, any two schemes labelled
with different letters behave significantly different, and this sub-process therefore can
be identified as sensitive. Otherwise, the sub-process is considered insensitive. For
simplicity, schemes of insensitive sub-process are not labeled. Moreover, schemes with
the letters late in the alphabet have smaller mean RMSEs and outperform the ones with
the letters forward in the alphabet. Using the two schemes in CRS process (hereafter
CRS(1) and CRS(2)) in Fig. 6 as an example. For the annual and warm season, CRS(1)
and CRS(2) were labeled with "B" and "A", respectively. In the cold season, none of
them were labeled with letters. As described above, the CRS process was sensitive for
SCEs simulation during the annual and warm season, and CRS(1) outperformed
CRS(2). However, it was not sensitive during the cold season.
Consistent with the influence degrees in Fig. 4, the performance difference
between schemes of the STC and SUB for SCEs simulation were significantly greater
than other processes. Most other physical processes showed significant but limited
difference. Schemes in BTR and TBOT processes, however, had no significant different
performance. Specifically, the performance order followed STC(1) > STC(2), SUB(2) >
SUB(1), SFC(2) > SFC(1), ALB(2) > ALB(1), CMB(2) > CMB(1) in both annual and

seasonal scales. RAD showed no obvious difference during the warm season, while RAD(3) outperformed RAD(1) and (2) during the cold season. For SNF, SNF(3) generally excel SNF(1) and SNF(2), especially during the warm season.

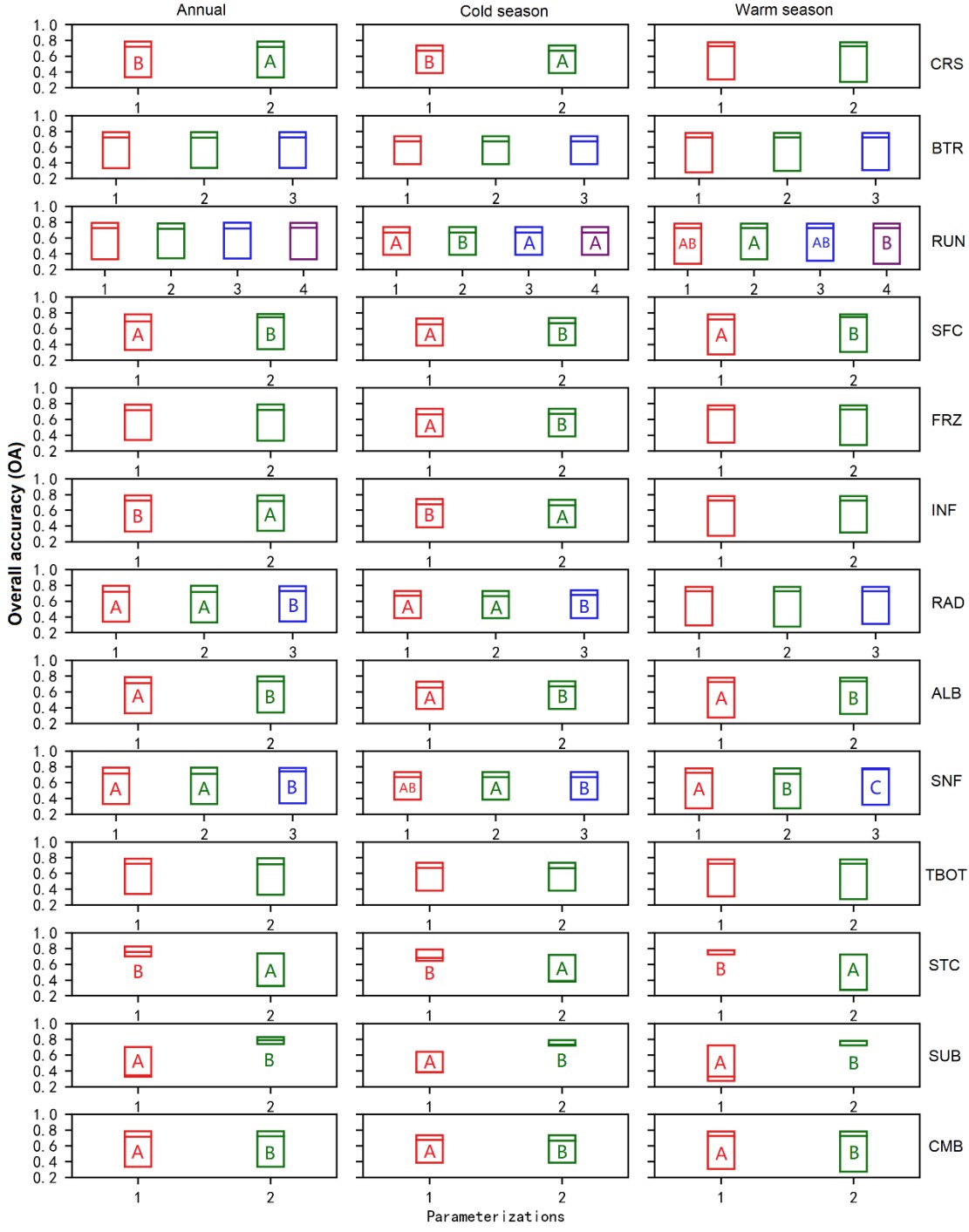

**Figure 6.** Distinction level for overall accuracy (OA) of snow cover events (SCEs) during the annual, warm, and cold seasons at TGL site. Limits of the boxes represent upper and lower quartiles, lines in the box indicate the median value.

All the physical processes showed sensitivities for ST and SLW simulation in

varying magnitudes except the BTR process and CRS process in most layers. For ST, the performance difference between schemes of the STC, SUB and SNF were obviously greater than other processes, indicating the importance of snow on ST, followed by the RAD, SFC and RUN processes. The performance orders followed STC(1) > STC(2), SUB(2) > SUB(1), SNF(3) > SNF(1) > SNF(2), RAD(3) > RAD(1) > RAD(2), and SFC(2) > SFC(1). For SLW, the RUN, STC, and SUB processes showed significant and higher sensitivities than other physical processes, especially during the warm season and at the shallow layers (Fig. xx). Consistent with that of ST, the performance orders for SLW simulation were STC(1) > STC(2), and SUB(2) > SUB(1).For the RUN process, the performance orders for both ST and SLW simulation generally followed RUN(4) > RUN(1) > RUN(3) > RUN(2) as a whole, among which RUN(1) and RUN(4) presented similar performance during both warm and cold seasons. During both warm and cold seasons, the performance orders for ST simulations were SFC(2) > SFC(1) for SFC process, FRZ(2) > FRZ(1) for FRZ process, and RAD(3) > RAD(1) > RAD(2) for RAD process (Fig. S2 and S3), which are particularly so for SLW simulations at shallow and deep layers.

For ST, both FRZ and INF showed higher sensitivities during the cold season, especially at shallow soils for FRZ and deep soils for INF. FRZ(2)/INF(1) outperformed FRZ(1)/INF(2) for the whole year for ST simulation. Specifically, FRZ(1)/INF(2) performed better at the shallow soils during the warm season while did worse during the cold season compared with FRZ(2)/INF(1). For SLW, FRZ(2)/INF(2) generally preceded FRZ(1)/INF(1) at shallow and deep soils (5 cm, 25 cm, 220 cm, and 300 cm) while did worse at middle soil layers (140 cm and 220 cm).

For ST simulation, the performance sequence in RAD and SNF was RAD(3) > RAD(1) > RAD(2) and SNF(3) > SNF (1) > SNF(2), respectively. For SLW simulation, the sequence become complicated. However, RAD(3) and RAD (3) still outperformed the other two schemes, respectively. ALB(2) was superior to ALB(1) for both ST and SLW simulation. The influence of TBOT on soil hydrothermal arose at deep soils and during cold season, and TBOT(1) excel TBOT (2). CMB(2) outperformed CMB(1) for

ST simulation, so did that for SLW simulation at shallow and deep soils (5 cm, 25 cm, and 300 cm).

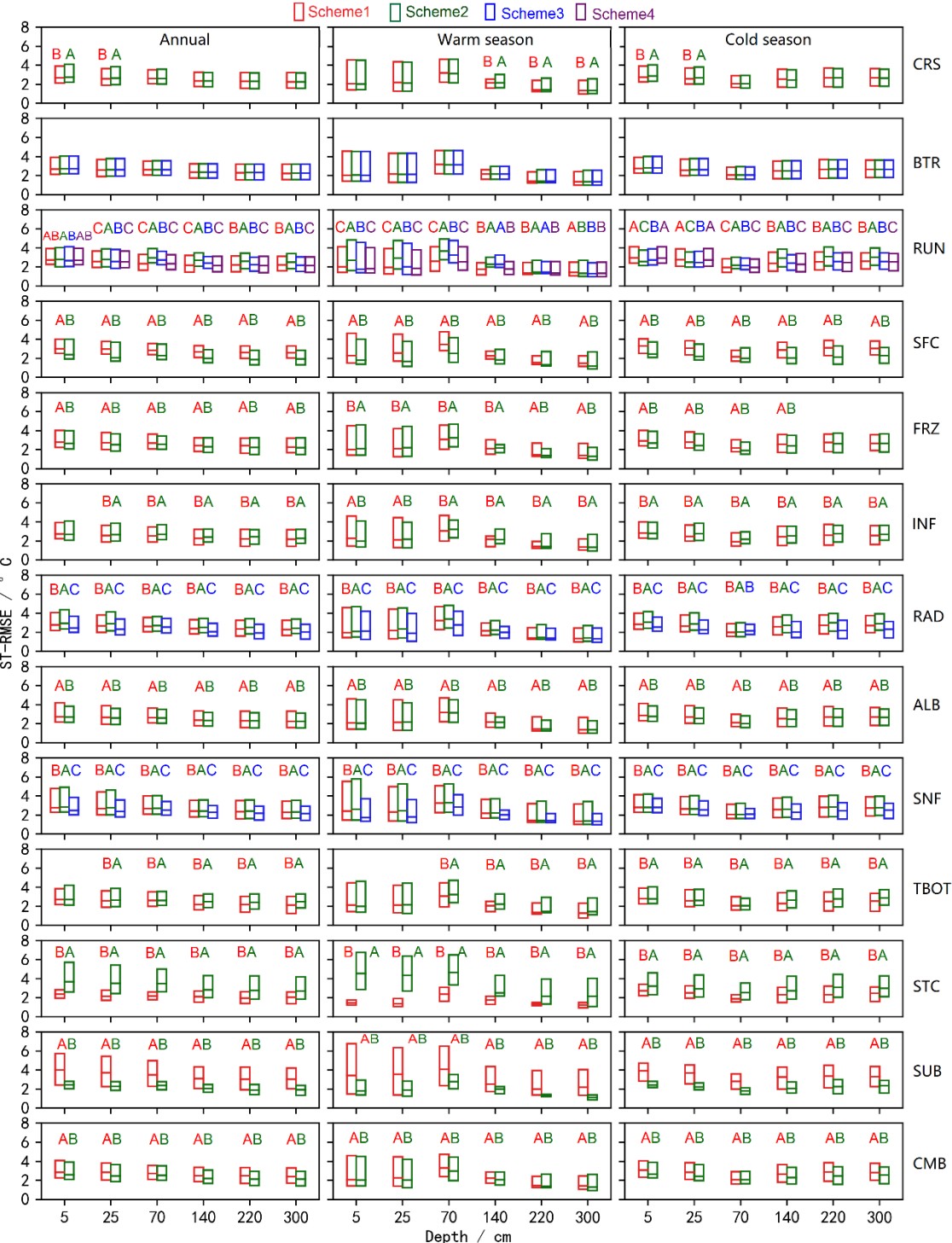

**Figure 7.** Distinction level for RMSE of ST at different layers during the annual, warm, and cold seasons in the ensemble simulations at TGL site. Limits of the boxes represent upper and lower quartiles, lines in the box indicate the median value.

362

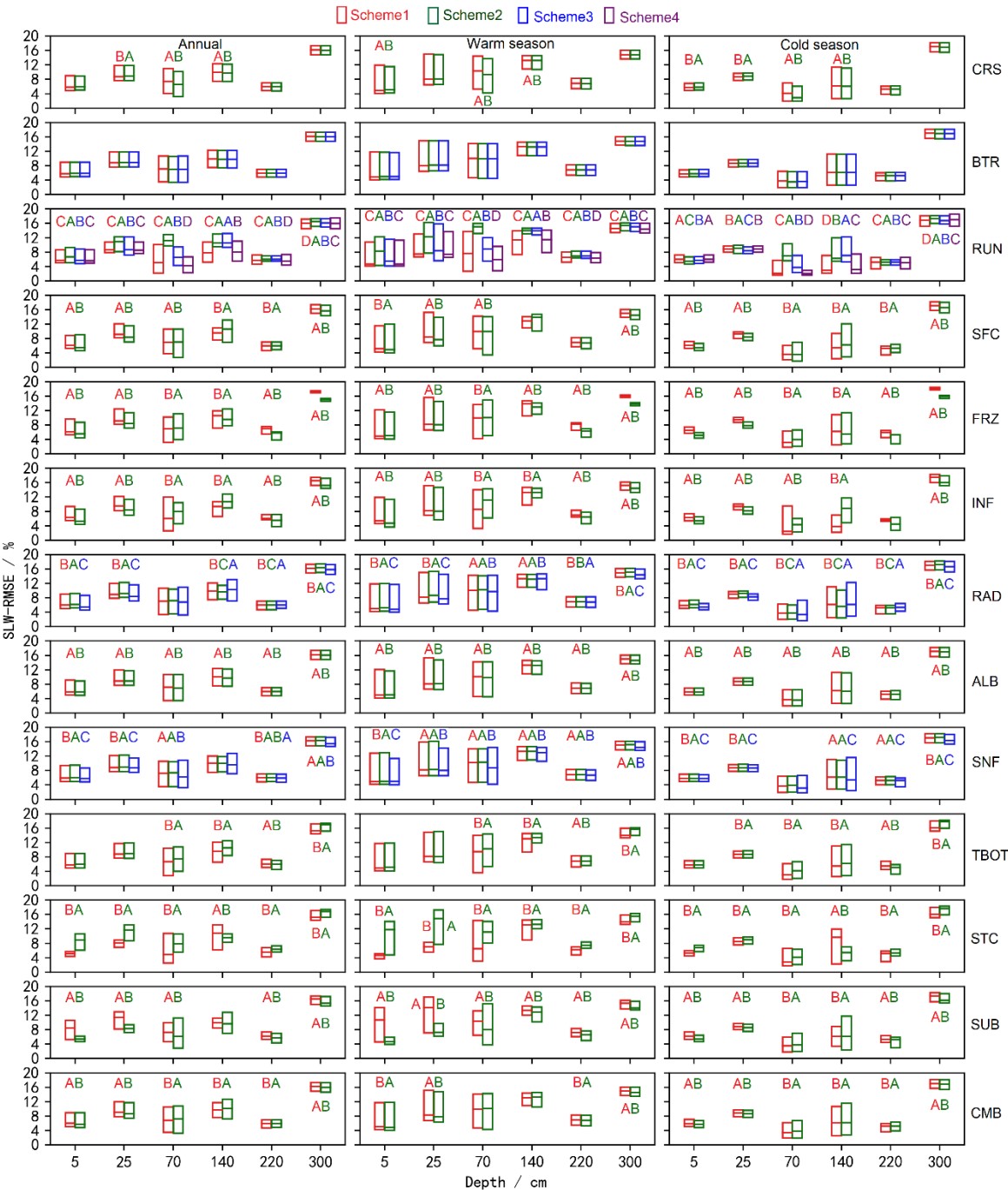

**Figure 8.** Same as in Figure 7 but for SLW.

## 3.3 Influence of snow cover and surface drag coefficient on soil hydrothermal

dynamics

367

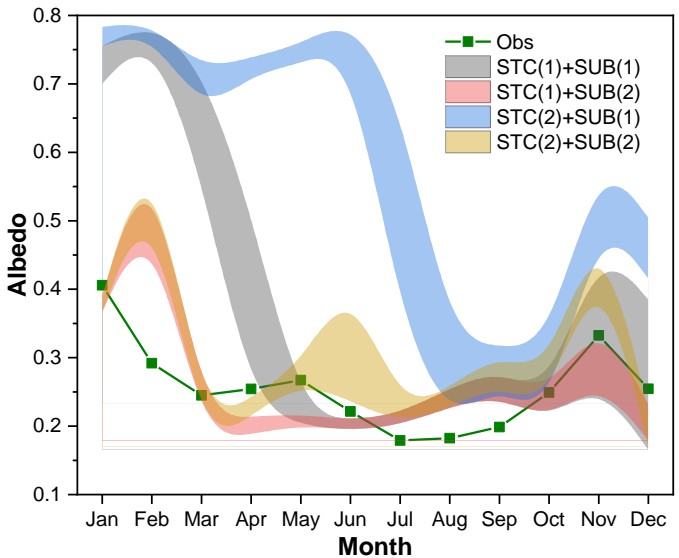

**Figure 9.** Uncertainty interval of ground albedo at TGL site in dominant physical processes (STC and SUB) for snow cover event simulation.

The influence of snow on soil temperature is firstly investigated. The dominant role of STC and SUB in the simulation of SCEs has been identified (Fig. 4 and 6). Interactions between the two physical processes are further analyzed here. Figure 9 compares the uncertainly intervals of the two physics. The duration of snow cover is the longest when STC(2)+SUB(1), followed when STC(2)+SUB(1). Simulations considering SUB(2) generally has a short snow duration. Among the four combinations, STC(1)+SUB(2) is in best agreement with the measurements.

Given the good performance of STC(1)+SUB(2) in simulating SCEs, the influence of snow on soil hydrothermal dynamics is investigated by comparing the total ensemble mean ST and SLW with those adopting STC(1)+SUB(2) (Fig. 3). It can be seen that the ensemble mean ST of simulations adopting STC(1) and SUB(2) are generally higher than the total ensemble means, especially during the spring and summer (Mar.-Aug.). In January and February at shallow layers (5 cm, 25 cm and 70 cm), STC(1)+SUB(2) had a lower ST and showed an insulation effect on ST during the two months. As a whole, however, snow cover has a cooling effect on ST. In addition, along with the improved SCEs and elevated ST, STC(1)+SUB(2) induced moister soil with higher SLW (Fig. 3).

389

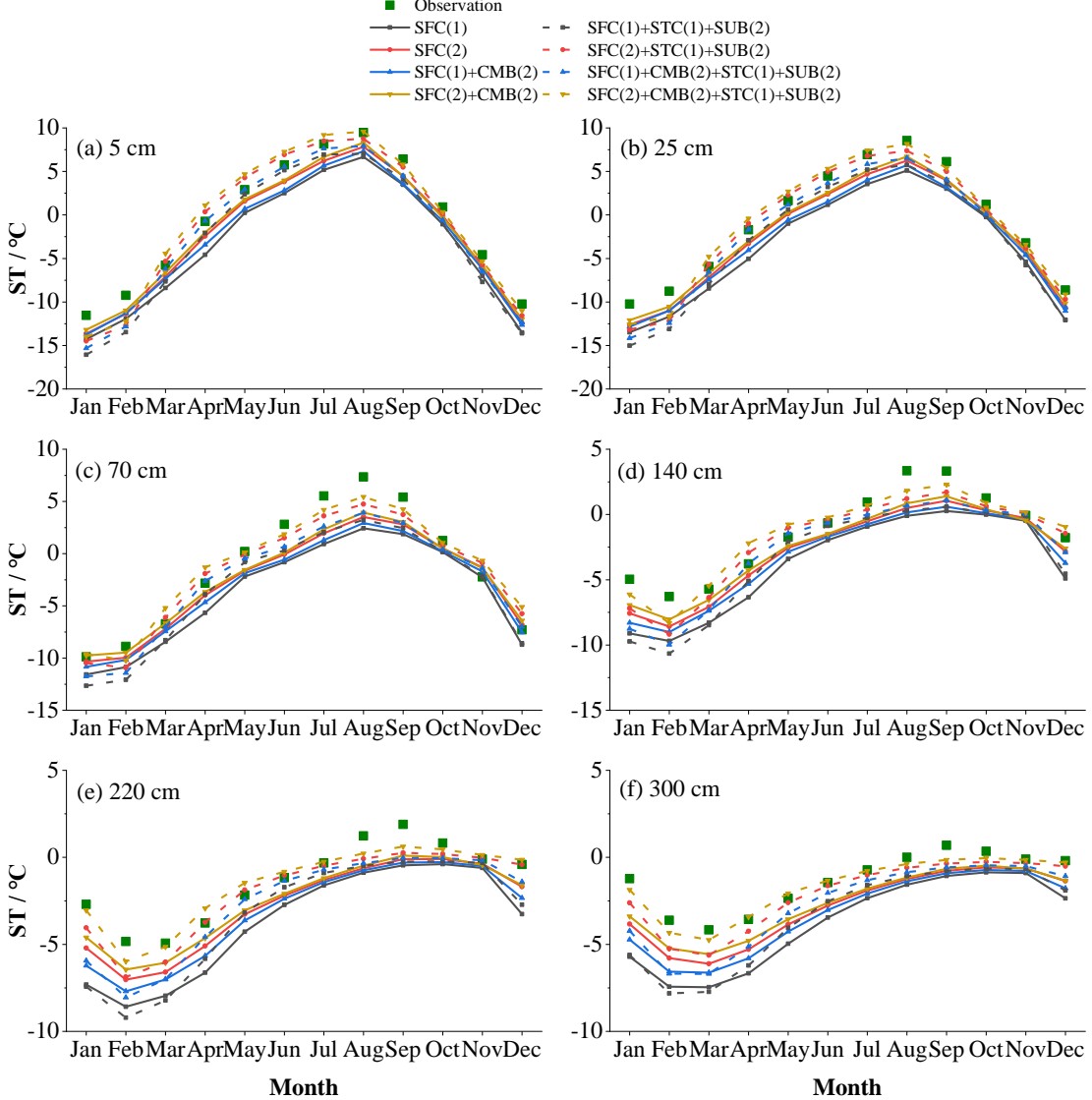

**Figure 10.** Monthly soil temperature (ST in °C) at (a) 5 cm, (b) 25 cm, (c) 70 cm, (d) 140 cm, (e) 220 cm, (f) 300 cm for the SFC process that consider the CMB(2) and STC(1)+SUB(2) processes or not.

SFC and CMB process using different ways to calculate the surface drag coefficient, which is of great influence for surface energy partitioning and thus ST and SLW. The influence of surface drag coefficient is assessed by comparing the soil temperature before and after considering the combined scheme (CMB(2)) and the effect of snow (STC(1)+SUB(2)) (Fig. 10). SFC(2) tended to produce higher ST than SFC(1), especially during the warming period (January-August). When adopting the combined scheme of Y08 and UCT (CMB(2)), the cold bias were significantly resolved. The

performance order followed SFC(2)+CMB(2) > SFC(2) > SFC(1)+CMB(2) > SFC(1).
However, considerable underestimations of ST still exist at all layers due to the poor
representation of snow process. After eliminating the effects of snow (STC(1)+SUB(2),
dash lines in Fig. 10), the simulated ST accordingly increased except in January and
February. SFC(2) and SFC(2)+CMB(2) overestimated STs from March to July at
shallow layers (5 cm and 25 cm), resulting in good agreements of deep STs with
observations. In contrast, the simulated STs at shallow layers (5 cm and 25 cm) by
SFC(1) and SFC(1)+CMB(2) were basically consistent with observations from March
to July. While large cold bias remained at deep layers.
**4   Discussion**
**4.1 Snow cover on the QTP and its influence on soil hydrothermal regime**
Snow cover in the permafrost regions of the QTP is thin, patchy, and short-lived
(Che et al., 2019), whose influence on soil temperature and permafrost state is usually
considered weak (Jin et al., 2008; Zou et al., 2017; Wu et al., 2018; Zhang et al., 2018;
Yao et al., 2019). However, our ensemble simulations showed that the surface albedo
is extremely overestimated in both magnitude and duration (Fig. 2), implying an
extreme overestimation of snow cover, which is consistent with the studies using Noah-
MP model (Jiang et al., 2020; Li et al., 2020; Wang et al., 2020) and widely found in
other state-of-the-art LSMs (Wei and Dong, 2015) on the QTP.
Great efforts to resolve the overestimation of snow cover in LSMs include
considering the vegetation effect (Park et al., 2016), the snow cover fraction (Jiang et
al., 2020), the blowing snow (Xie et al., 2019), and the fresh snow albedo (Wang et al.
2020). Our results illustrated the superiority of considering the snow sublimation from
wind (SUB(2)) and using semi-implicit snow/soil temperature time scheme (STC(1))
(Fig. 4, 6 and 9) when simulating snow cover on the QTP. It is consistent with previous
conclusions that accounting for the loss resulting from wind contributes to improve
snow cover days and depth (Yuan et al., 2016), and that STC(1) has a rapid snow
ablation than STC(2) (You et al., 2020a).
The impacts of snow cover on soil temperature in magnitude and vector (cooling or
warming) depend on its timing, duration, and depth (Zhang et al., 2005). In January and
February, the ground heat flux mainly goes upward, the warming effect of simulated
snow can be related to the overestimated snow depth that prevent heat loss from the
ground. During the spring and summer when snow melts, the cooling effects occurs,
mainly because considerable energy that used to heat the ground is reflected due to the
high albedo of snow. With the improvement of snow (STC(1)+SUB(2)), the originally
overestimated snow melts and infiltrated into the soil, resulting in improved SLWs (Fig.
3). And higher soil temperature also contributed to the SLWs according to the freezing-
point depression equation, in which SLW exponentially increase with soil temperature
for a given site (Niu and Yang, 2006).

## 4.2 Discussions on the sensitivity of physical processes on soil hydrothermal simulation

### 4.2.1 Canopy stomatal resistance (CRS) and soil moisture factor for stomatal resistance (BTR)

The biophysical process BTR and CRS directly affect the canopy stomatal
resistance and thus the plant transpiration (Niu et al., 2011). The transpiration of plants
could impact the ST/SLW through its cooling effect (Shen et al., 2015) and the water
balance of root zone (Chang et al., 2020). However, the annual transpiration of alpine
steppe is weak due to the shallow effective root zone and lower stomatal control in this
dry environment (Ma et al., 2015), which may explain the indistinctive or very small
difference among the schemes of the BTR and CRS processes for SCEs (Fig. 8), ST
(Fig. 7) and SLW (Fig. 8).

### 4.2.2 Runoff and groundwater (RUN)

In the warm season, different SLWs would result in the difference of the surface
energy partitioning and thus different soil temperatures. RUN(2) had the worst

performance for simulating ST and SLW (Fig. 7 and 8) among the four schemes, likely due to its higher estimation of soil moisture (Fig. S1) and thus greater sensible heat and smaller ST (Gao et al., 2015). Likewise, RUN(4) was on a par with RUN(1) in the simulation of ST at most layers due to the very small difference in SLW of two schemes (Fig. 8 and S1). For the whole soil column, RUN(4) surpassed RUN(1) and RUN(2) for SLW simulation, both of which define surface/subsurface runoff as functions of groundwater table depth (Niu et al., 2005; Niu et al., 2007). This is in keeping with the study of Zheng et al. (2017) that soil water storage-based parameterizations outperform the groundwater table-based parameterizations in simulating the total runoff in a seasonally frozen and high-altitude Tibetan river. Besides, RUN(4) is designed based on the infiltration-excess runoff (Yang and Dickinson, 1996) in spite of the saturation-excess runoff in RUN(1) and RUN(2) (Gan et al., 2019), which is more common in arid and semiarid areas like the permafrost regions of QTP (Pilgrim et al., 1988). In the cold season, much of the liquid water freezes into ice, which would greatly influence the thermal conductivity of frozen soil considering thermal conductivity of ice is nearly four times that of the equivalent liquid water. Therefore, the impact of RUN is important for the soil temperature simulations at both warm and cold seasons (Fig. 5 and 7).

**4.2.3 Surface layer drag coefficient (SFC and CMB)**

SFC defines the calculations of the surface exchange coefficient for heat and water vapor (CH), which greatly impact the energy and water balance and thus the temperature and moisture of soil (Zeng et al., 2012; Zheng et al., 2012). SFC(1) adopts the Monin-Obukhov similarity theory (MOST) with a general form, while the SFC(2) uses the improved MOST modified by Chen et al. (1997). In SFC(1), the roughness length for heat ($Z_{0h}$) is taken as the same with the roughness length for momentum ($Z_{0m}$, Niu et al., 2011). SFC(2) adopts the Zilitinkevitch approach for $Z_{0,h}$ calculation (Zilitinkevich, 1995). The difference between SFC(1) and SFC(2) has a great impact on the CH value. Several studies have reported that SFC(2) has a better performance for the simulation of sensible and latent heat on the QTP (Zhang et al., 2016; Gan et al., 2019). The results of T-test in this study showed remarkable distinctions between the

two schemes, where SFC(2) was dramatically superior to SFC(1) (Fig. 7 and 8). SFC(2) produces lower CH than SFC(1) (Zhang et al., 2014), resulting in less efficient ventilation and greater heating of the land surface (Yang et al., 2011b), and substantial improvement of the cold bias of Noah-MP in this study (Fig. 7 and 10).

Both SFC(1) and SFC(2) couldn't produce the diurnal variation of $Z_{0,h}$ (Chen et al., 2010). CMB offers a scheme that considered the diurnal variation of $Z_{0,h}$ in bare ground and under-canopy turbulent exchange in sparse vegetated surfaces (Li et al., 2020). Consistent with previous studies in the QTP (Chen et al., 2010; Guo et al., 2011; Zheng et al., 2015; Li et al., 2020), the simulated ST generally followed SFC(2)+CMB(2) > SFC(2) > SFC(1)+CMB(2) > SFC(1) with/without removing the overestimation of snow (Fig. 10), indicating that CMB(2) contributes to resolve the cold bias of LSMs. However, none of the four combinations could well reproduce the shallow and deep STs simultaneously. When the snow is well-simulated, SFC(2)+CMB(2) performed the best at deep layers at the cost of overestimating shallow STs. Meanwhile, SFC(1)+CMB(1) showed the best agreements at shallow layers with considerable cold bias at deep layers, which can be related to the overestimated frozen soil thermal conductivity (Luo et al., 2009; Chen et al., 2012; Li et al., 2019).

**4.2.4 Super-cooled liquid water (FRZ) and frozen soil permeability (INF)**

FRZ and INF describe the unfrozen water and permeability of frozen soil, and had a larger influence on ST/SLW during the cold season than warm season as expected (Fig. 5). Specifically, FRZ treats liquid water in frozen soil (super-cooled liquid water) using two forms of freezing-point depression equation. FRZ(1) takes a general form (Niu and Yang, 2006), while FRZ(2) exhibits a variant form that considers the increased surface area of icy soil particles (Koren et al., 1999). FRZ(2) generally yields more liquid water in comparison of FRZ(1) (Fig. S2). INF(1) uses soil moisture (Niu and Yang, 2006) while INF(2) employs only the liquid water (Koren et al., 1999) to parameterize soil hydraulic properties. INF(2) generally produces more impermeable frozen soil than INF(1), which is also found in this study (Fig. S3). For the whole year, INF(1) surpassed INF(2) in simulating STs, which may be related to the more realistic

SLWs produced by INF(1) for the whole soil column (Fig. S3).

**4.2.5 Canopy gap for radiation transfer (RAD)**

RAD treats the radiation transfer process within the vegetation, and adopts three
methods to calculate the canopy gap. RAD(1) defines canopy gap as a function of the
3D vegetation structure and the solar zenith angle, RAD(2) employs no gap within
canopy, and RAD(3) treat the canopy gap from unity minus the FVEG (Niu and Yang,
2004). The RAD(3) scheme penetrates the most solar radiation to the ground, followed
by the RAD(1) and RAD(2) schemes. As an alpine grassland, there is a relative low
LAI at TGL site, and thus a quite high canopy gap. So, schemes with a larger canopy
gap could realistically reflect the environment. Consequently, the performance
decreased in the order of RAD(3) > RAD(1) > RAD(2) for ST/SLW simulation.

**4.2.6 Snow surface albedo (ALB) and precipitation partition (SNF)**

The ALB describe two ways for calculating snow surface albedo, in which the
ALB(1) and ALB(2) adopt the scheme from BATS and CLASS LSM, respectively.
ALB(2) generally produce lower albedo than ALB(1), especially when the ground
covered by snow (Fig. S4). As a result, higher net radiation absorbed by the land surface
and more heat is available for heating the soil in ALB(2), which is beneficial for
counteracting the cooling effect of overestimated snow on ST (Fig. S5). Along with the
higher ST, ALB(2) outperformed ALB(1) for SLW simulation, likely due to more snow
melt water offset the dry bias in Noah-MP (Fig. S5).
The SNF defines the snowfall fraction of precipitation as a function of surface air
temperature. SNF(1) is the most complicated of the three schemes, in which the
precipitation is considered rain/snow when the surface air temperature is greater/less
than or equal to 2.5/0.5 °C, otherwise, it is recognized as sleet. While SNF(2) and
SNF(3) simply distinguish rain or snow by judging whether the air temperature is above
2.2 °C and 0 °C or not. The significant difference between three schemes for SCEs
simulation during the warm season is consistent with the large difference of snowfall
fraction in this period (Fig. 6 and S6). SNF(3) is the most rigorous scheme and produce
the minimum amount of snow, followed by SNF(1) and SNF(2) with limited difference
(Fig. S6). This exactly explains superiority of SNF(3) for ST and SLW simulation (Fig.
7 and 8).

**4.2.7 Lower boundary of soil temperature (TBOT) and snow/soil temperature time**

**scheme (STC)**

TBOT process adopts two schemes to describe the soil temperature boundary
conditions. TBOT (1) assumes zero heat flux at the bottom of the model, while TBOT(2)
adopts the soil temperature at the 8 m depth (Yang et al., 2011a). In general, TBOT(1)
is expected to accumulate heat in the deep soil and produce higher ST than TBOT(2).
In this study, the two assumptions performed significantly different, especially at the
deep soils and during the cold season. Although TBOT(2) is more representative of the
realistic condition, TBOT(1) surpassed TBOT(2) in this study. It can be related to the
overall underestimation of the model, which can be alleviated by TBOT(1) because of
heat accumulation (Fig. S7).
Two time discretization strategies are implemented in the STC process, where
STC(1) adopts the semi-implicit scheme while STC(2) uses the full implicit scheme, to
solve the thermal diffusion equation in first soil or snow layers (Yang et al., 2011a).
STC(1) and STC(2) are not strictly a physical processes but different upper boundary
conditions of soil column (You et al., 2020a). The differences between STC(1) and
STC(2) were significant (Fig. 7). The impacts of the two options on ST is remarkable
(Fig. 6), particularly in the shallow layers and during the warm season (Fig. 5). In
addition, STC(1) outperformed STC(2) in the ensemble simulated ST(Fig. 7), because
STC(1) greatly alleviated the cold bias in Noah-MP (Fig. S8) by producing the higher
OA of SCEs (Fig. 6)

**4.3 Perspectives**

This study analyzed the characteristics and general behaviors of each
parameterization scheme of Noah-MP at a typical permafrost site on the QTP, hoping
to provide a reference for simulating permafrost state on the QTP. We identified the
systematic overestimation of snow cover, cold bias and dry bias in Noah-MP, and
discussed the role of snow and surface drag coefficient on soil hydrothermal dynamics.
Further tests at another permafrost site (BLH site, 34.82° N, 92.92° E, Alt.: 4,659 m
a.s.l) basically showed consistent conclusions with that at TGL site (see Supplementary
files for details), indicating that relevant results and methodologies can be practical
guidelines for improving the parameterizations of physical processes and testing their
uncertainties towards soil hydrothermal modeling in the permafrost regions of the
plateau. Although the site we selected may be representative for the typical environment
on the plateau, continued investigation with a broad spectrum of climate and
environmental conditions is required to make a general conclusion at regional scale.
**5  Conclusions**
An ensemble simulation using multi-parameterizations was conducted using the
Noah-MP model at the TGL site, aiming to present a reference for simulating soil
hydrothermal dynamics in the permafrost regions of QTP using LSMs. The model was
modified to consider the vertical heterogeneity in the soil and the simulation depth was
extended to cover the whole active layer. The ensemble simulation consists of 55296
experiments, combining thirteen physical processes (CRS, BTR, RUN, SFC, FRZ, INF,
RAD, ALB, SNF, TBOT, STC, SUB, and CMB) each with multiple optional schemes.
On this basis, the general performance of Noah-MP was assessed by comparing
simulation results with in situ observations, and the sensitivity of snow cover event, soil
temperature and moisture at different depths of active layer to parameterization
schemes was explored. The main conclusions are as follows:
(1) Noah-MP model tends to overestimate snow cover, which is most influenced by the
STC and SUB processes. Such overestimation can be greatly resolved by
considering the snow sublimation from wind (SUB(2)) and semi-implicit snow/soil
temperature time scheme (STC(1)).
(2) Soil temperature is largely underestimated by the overestimated snow cover and
thus dominated by the STC and SUB processes. Systematic cold bias and large

uncertainties of soil temperature still exist after eliminating the effects of snow, particularly at the deep layers and during the cold season. The combination of Y08 and UCT contributes to resolve the cold bias of soil temperature.

(3) Noah-MP tend to underestimate soil liquid water content. Most physical processes have limited influence on soil liquid water content, among which the RUN process plays a dominant role during the whole year. The STC and SUB process have a considerable influence on topsoil liquid water during the warm season.

*Code availability.* The original source code of the offline 1D Noah-MP LSM v1.1 is available at

https://ral.ucar.edu/solutions/products/noah-multiparameterization-land-surface-model-noah-mp-lsm (last access: 23 February 2021). The modified Noah-MP with the consideration of vertical heterogeneity in soil profile, snow sublimation from wind and the combination of roughness length for heat and under-canopy aerodynamic resistance can be downloaded at http://doi.org/10.5281/zenodo.4555449.

*Data availability.* The 1-hourly forcing data, daily soil temperature and liquid water content at the TGL and BLH sites are available at https://doi.org/10.17632/h7hbd69nnr.2. Soil texture data can be obtained at https://doi.org/10.1016/j.catena.2017.04.011 (Hu et al., 2017). The AVHRR LAI data can be downloaded from https://www.ncei.noaa.gov/data/ (Claverie et al., 2016).

*Author contributions.* TW and XL conceived the idea and designed the model experiments. XL performed the simulations, analyzed the output, and wrote the paper. JC helped to compile the model in a GNU/Linux (CentOS 7.0) environment. XW, XZ, GH, RL contributed to the conduction of the simulation and interpretation of the results. YQ provided the observations of atmospheric forcing and soil temperature. CY and JH helped in downloading and processing the AVHRR LAI data. JN and WM provide guidelines for the visualization. Everyone revised and polished the paper.


*Competing interests.* The authors declare that they have no conflict of interest.

*Acknowledgements.* This work has been supported by the CAS "Light of West China"
Program, the National Natural Science Foundation of China (41690142; 41771076;
41961144021; 42071093), the CAS "Hundred Talents" Program (Sizhong Yang), and
the National Cryosphere Desert Data Center Program (E0510104). The authors thank
Cryosphere Research Station on the Qinghai-Tibet Plateau, CAS for providing field
observation data and Mr. Guohui Zhao for awarding us access to supercomputing
resources. We would like to thank Dr. Sizhong Yang and two anonymous reviewers for
their insightful and constructive comments and suggestions, which greatly improved
the quality of the manuscript.

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
