# Peer review of "Assessing the simulated soil hydrothermal regime of active layer"

_Geoscientific Model Development, 2020_

## Author Comment (AC1) · 25 Aug 2020

The authors found several typos:

1. All the "standard deviation/SD/SDs" in the original manuscript should be "mean bias error/MBE/MBEs".

2. Line 426: Tukey's test -> T-test

3. ". . . Noah-MPLSM v1.1 . . ." in the title should be ". . . Noah-MP LSM v1.1 . . ."

[Figure]

2020.

---

## Referee Comment (RC1) · Anonymous Referee #1 · 15 Sep 2020

The authors systematically evaluated the effects of different physical processes and associated parameterization options on Noah-MP simulated soil temperature at a permafrost site over the Tibetan Plateau. The manuscript is generally well-written and well-structured. Before it can be considered for potential publication, I have a few comments for the authors to consider.

Major comment:

1. I am not convinced why the authors did not test the snow-related processes and parameterizations, such as snow albedo and rain-snow partitioning schemes. These processes along with the snow cover formulation in Noah-MP will affect surface heat fluxes and energy balance, which can potentially affect soil temperature evolution below snowpack. Particularly, the authors found that Noah-MP generally underestimates the soil temperature during the cold season, which could partially be related to snowpack simulations. The authors also did not tell the readers that what parameterization schemes they used for snow albedo and partitioning processes. Moreover, a recent study over Tibetan Plateau (Jiang et al., 2020, https://agupubs.onlinelibrary.wiley.com/doi/abs/10.1029/2020JD032674) showed that the processes already tested by the authors here along with the snow cover formulation can significantly affect snowpack simulations, which could further affect soil conditions. Thus, it is likely that the processes the authors tested can indirectly affect soil conditions through modifying snowpack. I suggest the authors add some discussions on this aspect and include some quick tests for snow-related processes if possible.

Minor comments:

1. Line 108: "depth" -> "depths".

2. Line 170: Please give some details on how the soil column was discretized, e.g., how many soil layers, the thickness of each layer, etc.

3. Line 189: What is "Si"?

4. What is the model timestep in the simulations in this study?

5. Section 4.3: The authors only tested the model performance at one site. So to what extent their conclusions can be extended to other Tibetan Plateau areas?

---

## Referee Comment (RC2) · Anonymous Referee #2 · 13 Oct 2020

It's my pleasure to review gmd-2020-142 "Assessing the simulated soil thermal regime from Noah-MP LSM v1.1 for near-surface permafrost modeling on the Qinghai-Tibet Plateau" by Li et al. The authors evaluate the performance of Noah-MP in simulating soil temperature on a permafrost site over the Tibetan Plateau. There are many additional work need to be done before this paper can be accepted. 1. I note that there is a paper recently published by the same author to improve the performance of Noah-MP simulations on the same site. It will be interesting the authors firstly add their improvements, and then design more numerical experiments to test the uncertainties of different parameterization options. Since one additional site, soil moisture and snow measurements are available, the authors are suggested to also use these measurements to test the Noah-MP's performance. For the frozen soil, the soil moisture and soil temperature are fully coupled, which are also affected by the snow process, so it's also important to evaluate the performance of Noah-MP in simulating these variables. 2. Since the snow process is also important for permafrost soil temperature simulations, it's suggested to also consider the impact of ALB and SNF options. 3. It's also suggested to evaluate the performance of Noah-MP for frozen (e.g. October-April) and thawed (e.g. May-September) soil conditions separately. Because it's very strange to me that the impact of RUN is so important for the soil temperature simulations. 4. Detailed information is needed for the following descriptions "The soil 164 hydraulic parameters, including the porosity, saturated hydraulic conductivity, hydraulic potential, the Clapp-Hornberger parameter b, field capacity, wilt point, and saturated soil water diffusivity, were determined using the pedotransfer functions proposed by Hillel (1980), Cosby et al. (1984), and Wetzel and Chang (1987), in which the sand and clay percentages were based on Hu et al., (2017). In addition, the simulation depth was extended to 8.0 m to cover the active layer thickness of the QTP. The soil column was discretized following the default scheme in CLM 5.0 (Lawrence et al., 2018)."

---

## Author Comment (AC2) · 3 Nov 2020

Please refer to the supplement

Please also note the supplement to this comment:
https://gmd.copernicus.org/preprints/gmd-2020-142/gmd-2020-142-AC2-supplement.pdf

---

## Author Comment (AC3) · 3 Nov 2020

We thank the reviewer for the insightful and constructive comments. We have made point-by-point responses and/or revisions according to your suggestions and instructions. We recall the comments of the reviewer in black, followed by our reply in blue.

Please note that we have rerun the simulations involving RUN(3) as replied to the comment #3 from referee #2, and all the figures in the manuscript have been revised accordingly.

The revised manuscript with tracking of all the changes that have been made is appended at the end of this response.

**Responses to Referee #1**

**Anonymous Referee #1**

The authors systematically evaluated the effects of different physical processes and associated parameterization options on Noah-MP simulated soil temperature at a permafrost site over the Tibetan Plateau. The manuscript is generally well-written and well-structured. Before it can be considered for potential publication, I have a few comments for the authors to consider.

Major comment:

1. I am not convinced why the authors did not test the snow-related processes and parameterizations, such as snow albedo and rain-snow partitioning schemes. These processes along with the snow cover formulation in Noah-MP will affect surface heat fluxes and energy balance, which can potentially affect soil temperature evolution below snowpack. Particularly, the authors found that Noah-MP generally underestimates the soil temperature during the cold season, which could partially be related to snowpack simulations. The authors also did not tell the readers that what parameterization schemes they used for snow albedo and partitioning processes.

Moreover, a recent study over Tibetan Plateau (Jiang et al., 2020, https://agupubs.onlinelibrary.wiley.com/doi/abs/10.1029/2020JD032674) showed that the processes already tested by the authors here along with the snow cover formulation can significantly affect snowpack simulations, which could further affect soil conditions. Thus, it is likely that the processes the authors tested can indirectly affect soil conditions through modifying snowpack. I suggest the authors add some discussions on this aspect and include some quick tests for snow-related processes if possible.

**Response:** Thank you for your constructive suggestion! In the revised manuscript, we have conducted an ensemble of 41472 (= 6912*2*3) experiments to test the performance of Noah-MP in simulating snow processes. Results show that Noah-MP extremely overestimates the albedo and thus induces great cold bias in soil temperature. Detailed results and discussions are illustrated in the newly added Sec. 3.1.1 and Sec. 4.1, respectively.

In addition, snow process is not considered by setting the snow fraction in precipitation to zero in this study. Since no snow cover in the ground, the ground albedo equals the soil albedo. We have added some explanations in lines 164-167: "For practical purpose, the ALB and SNF processes were not considered by setting the snow fraction in precipitation to zero. Since no snow cover in the ground, the ground albedo equals the soil albedo".

Minor comments:

1. Line 108: "depth" -> "depths".

**Response:** Revised as suggested.

2. Line 170: Please give some details on how the soil column was discretized, e.g., how many soil layers, the thickness of each layer, etc.

**Response:** The details of each layer are listed in the supplementary file as Table S1:

**Table S1** Soil discretization scheme and soil particle fraction in this study.

| Layer | $Z_i$ | $\Delta Z_i$ | $Z_{h,i}$ | Sand (%) | Silt (%) | Clay (%) |
|-------|-------|--------------|-----------|----------|----------|----------|

| | | | | | | |
|---|---|---|---|---|---|---|
| 1 | 0.010 | 0.020 | 0.020 | | | |
| 2 | 0.040 | 0.040 | 0.060 | 85.48 | 12.59 | 1.93 |
| 3 | 0.090 | 0.060 | 0.120 | | | |
| 4 | 0.160 | 0.080 | 0.200 | 83.51 | 13.57 | 2.92 |
| 5 | 0.260 | 0.120 | 0.320 | 81.15 | 15.58 | 3.27 |
| 6 | 0.400 | 0.160 | 0.480 | 86.62 | 11.16 | 2.22 |
| 7 | 0.580 | 0.200 | 0.680 | 78.73 | 18.06 | 3.21 |
| 8 | 0.800 | 0.240 | 0.920 | 88.12 | 8.98 | 2.90 |
| 9 | 1.060 | 0.280 | 1.200 | 95.00 | 3.00 | 2.00 |
| 10 | 1.360 | 0.320 | 1.520 | | | |
| 11 | 1.700 | 0.360 | 1.880 | 92.50 | 4.00 | 3.50 |
| 12 | 2.080 | 0.400 | 2.280 | 90.00 | 5.00 | 5.00 |
| 13 | 2.500 | 0.440 | 2.720 | | | |
| 14 | 2.990 | 0.540 | 3.260 | | | |
| 15 | 3.580 | 0.640 | 3.900 | | | |
| 16 | 4.270 | 0.740 | 4.640 | 68.00 | 20.00 | 12.00 |
| 17 | 5.060 | 0.840 | 5.480 | | | |
| 18 | 5.950 | 0.940 | 6.420 | | | |
| 19 | 6.940 | 1.040 | 7.460 | | | |
| 20 | 7.980 | 1.040 | 8.500 | | | |

Layer node depth ($Z_i$), thickness ($\Delta Z_i$), and depth at layer interface ($Z_{h,i}$) for default soil column. All in meters.

Accordingly, we revised the sentences in lines 174-186 as "The soil hydraulic parameters, including the porosity, saturated hydraulic conductivity, hydraulic potential, the Clapp-Hornberger parameter b, field capacity, wilt point, and saturated soil water diffusivity, were determined using the pedotransfer functions proposed by Hillel (1980), Cosby et al. (1984), and Wetzel and Chang (1987) (Equations S1-S7), in which the sand and clay percentages were based on Hu et al., (2017) (Table S1). In addition, the simulation depth was extended to 8.0 m to cover the active layer thickness of the QTP. The soil column was discretized into 20 layers, whose depths follow the default scheme in CLM 5.0 (Table S1, Lawrence et al., 2018). Due to the inexact match between observed and simulated depths, the simulations at 4cm, 26cm, 80cm, 136cm, 208cm and 299cm were compared with the observations at 5cm, 25cm, 70cm, 140cm, 220cm and 300cm, respectively. A 30-year spin-up was conducted in every simulation to reach equilibrium soil states.".

3. Line 189: What is "Si"?

**Response:** Sorry for the typo. It should be $\Delta\overline{RMSE}$, which has been corrected in line 203.

4. What is the model timestep in the simulations in this study?

**Response:** The model was driven by 1-hr-interval atmospheric forcing data, which has been described in lines 138-144: "The atmospheric forcing data, including wind speed/direction, air temperature/relative humidity/pressure, downward shortwave/longwave radiation, and precipitation, were used to drive the model. These variables above were measured at a height of 2 m and covered the period from August 10, 2010 to August 10, 2012 (Beijing time) with a temporal resolution of 1 hour. Daily soil temperature and moisture at depths of 5cm, 25cm, 70cm, 140cm, 220cm and 300cm from October 1, 2010 to September 30, 2011 (Beijing time) were utilized to validate the simulation results."

5. Section 4.3: The authors only tested the model performance at one site. So to what extent their conclusions can be extended to other Tibetan Plateau areas?

**Response:** Thanks for this review. We agree that further work is required in the future as discussed in Sec. 4.4. In this study, our main goal is to provide a reference for simulating permafrost state on the Tibet Plateau. However, before the whole Tibetan Plateau can be investigated, it is necessary to conduct such study at the site scale.

We believe the conclusion of the cold bias of Noah-MP in the Tibetan Plateau and the possible reasons are of high reliability. The study site is a typical permafrost site on the plateau with semiarid climate (Li et al., 2019), filmy and discontinuous snow cover (Che et al., 2019), sparse grassland (Yao et al., 2011), coarse soil (Wu and Nan, 2016; He et al., 2019), and thick active layer (Luo et al., 2016), which are common features in the permafrost regions of the plateau. In addition, such underestimations and the inabilities of producing the snow depth, diurnal $Z_{0h}$ and frozen soil thermal conductivity are widely reported in many state-of-the-art land surface models as discussed in Sec. 4.1 and 4.2.

In addition, the sensitivity analysis and optimal configuration of the physical processes in this study could contribute to better understand the land surface processes and provide practical guidelines for permafrost modeling at least in the permafrost areas with similar conditions on the plateau. Relevant methodologies could be generalized to other regions using the proposed approaches.

To be more unbiased and objective, we added some descriptions about the study site, and the new version in lines 126-131 are: "Tanggula observation station (TGL) lies in the continuous permafrost regions of Tanggula Mountain, central QTP (33.07°N, 91.93°E, Alt.: 5,100 m a.s.l; Fig. 1). This site a typical permafrost site on the plateau with sub-frigid and semiarid climate (Li et al., 2019), filmy and discontinuous snow cover (Che et al., 2019), sparse grassland (Yao et al., 2011), coarse soil (Wu and Nan, 2016; He et al., 2019), and thick active layer (Luo et al., 2016), which are common features in the permafrost regions of the plateau.".

And the perspective part (section 4.4) in lines 603-612 are rephrased as: "This study analyzed the characteristics and general behaviors of each parameterization scheme of Noah-MP at a typical permafrost site on the QTP, hoping to provide a reference for simulating permafrost state on the QTP. We identified the systematic overestimation of snow cover and cold bias in Noah-MP, and discussed the possible sources of error. Relevant results and methodologies can be practical guidelines for improving the parameterizations of physical processes and testing their uncertainties towards near-surface permafrost modeling on the plateau. Although the site we selected may be representative for the typical environment on the plateau, continued investigation with a broad spectrum of climate and environmental conditions is required to make a general conclusion at regional scale.".

**Other changes:**

- Thanks to the funded projects and referees in lines 683-688: "This work has been supported by the CAS "Light of West China" Program, and the National Natural Science Foundation of China (41690142; 41771076; 41961144021; 41671070). The authors thank Cryosphere Research Station on the Qinghai-Tibet Plateau, CAS for providing field observation data used in this study. We would like to thank two anonymous reviewers for their insightful and constructive comments and suggestions, which greatly improved the quality of the manuscript."
- We have rerun the simulations involving RUN(3) as replied to the comment #3 from referee #2, and all the figures in the manuscript have been revised accordingly.
- All the unfrozen water in the manuscript have been revised as soil liquid water (SLW).
- Delete "under review" in line 161
- Rewrite the sentences in lines 193-196 as: "The root mean square error (RMSE) between the simulations and observations were adopted to evaluate the performance of Noah-MP. The average of the RMSEs of all the soil layers was defined as column RMSE (colRMSE)."
- The study of Li et al. (2015) is cited in line 200:

Li, K., Gao, Y., Fei, C., Xu, J., Jiang, Y., Xiao, L., Li, R., and Pan, Y.: Simulation of impact of roots on soil moisture and surface fluxes over central Qinghai − Xizang Plateau. Plateau Meteor., 34, 642-652, https://doi.org/10.7522/j.issn.1000-0534.2015.00035, 2015.

- Delete the interaction analysis part in lines 328-346


**Response:** Thanks for this comment. The recently published work you mentioned only tested and augmented one selected combination of Noah-MP options. However, this study investigated the general performance and sensitivity of original Noah-MP model with all possible combinations, hoping to provide a reference for simulating permafrost state on the Tibet Plateau. The augmentation work is another big issue and out of scope of this paper. We choose not to add the suggested experiments, but highlight the continued efforts to augment the parameterizations of physical processes and test their uncertainties in the future in lines 603-612:

"This study analyzed the characteristics and general behaviors of each parameterization scheme of Noah-MP at a typical permafrost site on the QTP, hoping to provide a reference for simulating permafrost state on the QTP. We identified the systematic overestimation of snow cover and cold bias in Noah-MP, and discussed the possible sources of error. Relevant results and methodologies can be practical guidelines for improving the parameterizations of physical processes and testing their uncertainties towards near-surface permafrost modeling on the plateau. Although the site we selected may be representative for the typical environment on the plateau, continued investigation with a broad spectrum of climate and environmental conditions is required to make a general conclusion at regional scale."

With these revisions, we believe the potential readers can understand that our study aims to test the performance of the original Noah-MP, while future work is needed at the plateau scale.

Since one additional site, soil moisture and snow measurements are available, the authors are suggested to also use these measurements to test the Noah-MP's performance. For the frozen soil, the soil moisture and soil temperature are fully coupled, which are also affected by the snow process, so it's also important to evaluate the performance of Noah-MP in simulating these variables.

**Response:** We agree that add more sites would strengthen our conclusions. However, we realized that this will make our manuscript very long, and it is difficult to descript the results due to the different environmental factors among the sites. Our main goal is to provide a reference for simulating permafrost state on the Tibet Plateau. We tried our best to make this manuscript concise. Therefore, we would rather focus on one site, and it would be easier for potential readers to understand the core ideas.

We realized that potential readers may wonder why we did not assess the model using more data. To be clear, we explained this in the revised version in lines 603-612 as follows: "This study analyzed the characteristics and general behaviors of each parameterization scheme of Noah-MP at a typical permafrost site on the QTP, hoping to provide a reference for simulating permafrost state on the QTP. We identified the systematic overestimation of snow cover and cold bias in Noah-MP, and discussed the possible sources of error. Relevant results and methodologies can be practical guidelines for improving the parameterizations of physical processes and testing their uncertainties towards near-surface permafrost modeling on the plateau. Although the site we selected may be representative for the typical environment on the plateau, continued investigation with a broad spectrum of climate and environmental conditions is required to make a general conclusion at regional scale.".

To be more unbiased and objective, we added more descriptions about the study site, in lines 126-131: "Tanggula observation station (TGL) lies in the continuous permafrost regions of Tanggula Mountain, central QTP (33.07°N, 91.93°E, Alt.: 5,100 m a.s.l; Fig. 1). This site a typical permafrost site on the plateau with sub-frigid and semiarid climate (Li et al., 2019), filmy and discontinuous snow cover (Che et al., 2019), sparse grassland (Yao et al., 2011), coarse soil (Wu and Nan, 2016; He et al., 2019), and thick active layer (Luo et al., 2016), which are common features in the permafrost regions of the plateau.".

With these revisions, we believe the potential readers can understand our main findings. We keep the manuscript not too lengthy.

● About snow
As the reply to Referee #1, we conducted 41472 simulations to test the performance of Noah-MP in simulating snow cover. Similar with the recently published paper you mentioned (Li et al., 2020), ground albedo was used to roughly reflect the snow events. Our results show that Noah-MP extremely overestimates the albedo and thus induces great cold bias in soil temperature. Detailed results and discussions are illustrated in the newly added Sec. 3.1.1 and Sec. 4.1, respectively.

● About soil moisture
We checked the performance of Noah-MP in simulating soil liquid water (SLW) in the revised manuscript. Results show that the Noah-MP model generally underestimates soil moisture across the profile. The RUN process dominates the SLW simulation in comparison of the very limited impacts of all other physical processes. Detailed results can be found in lines Sec. 3.1.2 and Sec. 3.2.2.

2. Since the snow process is also important for permafrost soil temperature simulations, it's suggested to also consider the impact of ALB and SNF options.

**Response:** In the revised manuscript, we firstly checked the performance of Noah-MP for snow simulation and its impacts on soil temperature by considering the ALB and SNF options. Results showed that Noah-MP greatly overestimates snow cover both in magnitude and duration, inducing huge cold bias and large uncertainties in soil temperatures. However, our in-situ measurements and other studies show that snow cover has a very limited influence on soil temperature. Given the poor simulation of Noah-MP for snow cover and the weak impact of snow on soil temperature in reality, we did not consider the snow process in the following parts.

Detailed results and discussions are illustrated in the newly added Sec. 3.1.1 and Sec. 4.1, respectively.

3. It's also suggested to evaluate the performance of Noah-MP for frozen (e.g. October-April) and thawed (e.g. May-September) soil conditions separately. Because it's very strange to me that the impact of RUN is so important for the soil temperature simulations.

**Response:** We firstly apologize for the wrong coding when modifying the default Noah-MP to consider the vertical heterogeneity in the soil profile. In the wrong version, the maximum infiltration rate in RUN(3) was calculated as a function of all the soil layers (up to 8m in this study). Due to the existence of permafrost below 3m at the study site, the calculated infiltration rate is extremely small, resulting in small soil moisture of RUN(3) (Figure S1 in previous manuscript) and thus great influence degree of RUN process (Figure 3 in previous manuscript).

Following the default Noah-MP, we have rewritten the infiltration rate in RUN(3) as a function of the soil layers no more than 2m. Based on this, we reassessed the performance of Noah-MP for frozen and thawed soil conditions.

However, the main conclusion is consistent with previous manuscript except the declined influence of RUN process on soil temperature simulation. We have rewritten the main conclusions in lines 640-659 as:

(1) "Noah-MP model tends to overestimate snow cover and thus largely underestimate soil temperature in the permafrost regions of the QTP. Systematic cold bias and large uncertainties of soil temperature still exist after removing the snow processes, particularly at the deep layers and during the cold season. This is largely due to the imperfect model structure with regard to the roughness length for heat and soil thermal conductivity.

(2) Soil temperature is dominated by the surface layer drag coefficient (SFC) while largely influenced by runoff and groundwater (RUN). Other physical processes have little impact on ST simulation, among which VEG, RAD, and STC are more influential on shallow ST, while FRZ, INF and TBOT have greater impacts on deep ST. In addition, CRS and BTR do not significantly affect the simulation results.

(3) The best scheme combination for permafrost simulation are as follows: VEG (table LAI, calculated vegetation fraction), CRS (Jarvis), BTR (Noah), RUN (BATS), SFC (Chen97), RAD (zero canopy gap), FRZ (variant freezing-point depression), INF (hydraulic parameters defined by soil moisture), TBOT (ST at 8 m), STC (semi-implicit)."

4.Detailed information is needed for the following descriptions "The soil 164 hydraulic parameters, including the porosity, saturated hydraulic conductivity, hydraulic potential, the Clapp-Hornberger parameter b, field capacity, wilt point, and saturated soil water diffusivity, were determined using the pedotransfer functions proposed by Hillel (1980), Cosby et al. (1984), and Wetzel and Chang (1987), in which the sand and clay percentages were based on Hu et al., (2017). In addition, the simulation depth was extended to 8.0 m to cover the active layer thickness of the QTP. The soil column was discretized following the default scheme in CLM 5.0 (Lawrence et al., 2018)."

**Response:** We have added the details of the pedotransfer functions, the discretization scheme of soil column, and the soil particle fractions in the supplementary file:

The soil hydraulic parameters of each layer, including the porosity ($\theta_s$), saturated hydraulic conductivity ($K_s$), hydraulic potential ($\psi_s$), the Clapp-Hornberger parameter ($b$), field capacity ($\theta_{ref}$), wilt point ($\theta_w$), and saturated soil water diffusivity ($D_s$), were determined using the pedotransfer functions proposed by Hillel (1980), Cosby et al. (1984), and Wetzel and Chang (1987):

$$\theta_s = 0.489 - 0.00126(\%sand) \tag{S1}$$

$$K_s = 7.0556 \times 10^{-6.884+0.0153(\%sand)} \tag{S2}$$

$$\psi_s = -0.01 \times 10^{1.88-0.0131(\%sand)} \tag{S3}$$

$$b = 2.91 + 0.159(\%clay) \tag{S4}$$

$$\theta_{ref} = \theta_s \left[ \frac{1}{3} + \frac{2}{3} \left( \frac{5.79 \times 10^{-9}}{K_s} \right)^{1/(2b+3)} \right] \tag{S5}$$

$$\theta_w = 0.5\theta_s \left( \frac{-200}{\psi_s} \right)^{-1/b} \tag{S6}$$

$$D_s = b \cdot K_s \cdot \left( \frac{\psi_s}{\theta_s} \right) \tag{S7}$$

where $\%sand$ and $\%clay$ represent the percentage (%) of sand and clay content in soil, respectively.

**Table S1** Soil discretization scheme and soil particle fraction in this study.

| Layer | $Z_i$ | $\Delta Z_i$ | $Z_{h,i}$ | Sand (%) | Silt (%) | Clay (%) |
|-------|-------|--------------|-----------|----------|----------|----------|
| 1 | 0.010 | 0.020 | 0.020 | | | |
| 2 | 0.040 | 0.040 | 0.060 | 85.48 | 12.59 | 1.93 |
| 3 | 0.090 | 0.060 | 0.120 | | | |
| 4 | 0.160 | 0.080 | 0.200 | 83.51 | 13.57 | 2.92 |
| 5 | 0.260 | 0.120 | 0.320 | 81.15 | 15.58 | 3.27 |
| 6 | 0.400 | 0.160 | 0.480 | 86.62 | 11.16 | 2.22 |
| 7 | 0.580 | 0.200 | 0.680 | 78.73 | 18.06 | 3.21 |
| 8 | 0.800 | 0.240 | 0.920 | 88.12 | 8.98 | 2.90 |
| 9 | 1.060 | 0.280 | 1.200 | 95.00 | 3.00 | 2.00 |
| 10 | 1.360 | 0.320 | 1.520 | | | |
| 11 | 1.700 | 0.360 | 1.880 | 92.50 | 4.00 | 3.50 |
| 12 | 2.080 | 0.400 | 2.280 | | | |
| 13 | 2.500 | 0.440 | 2.720 | 90.00 | 5.00 | 5.00 |
| 14 | 2.990 | 0.540 | 3.260 | | | |
| 15 | 3.580 | 0.640 | 3.900 | | | |
| 16 | 4.270 | 0.740 | 4.640 | 68.00 | 20.00 | 12.00 |
| 17 | 5.060 | 0.840 | 5.480 | | | |

| 18 | 5.950 | 0.940 | 6.420 | | | |
| 19 | 6.940 | 1.040 | 7.460 | | | |
| 20 | 7.980 | 1.040 | 8.500 | | | |

Layer node depth ($Z_i$), thickness ($\Delta Z_i$ ), and depth at layer interface ($Z_{h,i}$) for default soil column. All in meters.

Accordingly, we revised the sentences in lines 174-186 as "The soil hydraulic parameters, including the porosity, saturated hydraulic conductivity, hydraulic potential, the Clapp-Hornberger parameter b, field capacity, wilt point, and saturated soil water diffusivity, were determined using the pedotransfer functions proposed by Hillel (1980), Cosby et al. (1984), and Wetzel and Chang (1987) (Equations S1-S7), in which the sand and clay percentages were based on Hu et al., (2017) (Table S1). In addition, the simulation depth was extended to 8.0 m to cover the active layer thickness of the QTP. The soil column was discretized into 20 layers, whose depths follow the default scheme in CLM 5.0 (Table S1, Lawrence et al., 2018). Due to the inexact match between observed and simulated depths, the simulations at 4cm, 26cm, 80cm, 136cm, 208cm and 299cm were compared with the observations at 5cm, 25cm, 70cm, 140cm, 220cm and 300cm, respectively. A 30-year spin-up was conducted in every simulation to reach equilibrium soil states.".

**Other changes:**

- Thanks to the funded projects and referees in lines 683-688: "This work has been supported by the CAS "Light of West China" Program, and the National Natural Science Foundation of China (41690142; 41771076; 41961144021; 41671070). The authors thank Cryosphere Research Station on the Qinghai-Tibet Plateau, CAS for providing field observation data used in this study. We would like to thank two anonymous reviewers for their insightful and constructive comments and suggestions, which greatly improved the quality of the manuscript."
- We have rerun the simulations involving RUN(3) as replied to the comment #3 from referee #2, and all the figures in the manuscript have been revised accordingly.
- All the unfrozen water in the manuscript have been revised as soil liquid water (SLW).
- Delete "under review" in line 161
- Rewrite the sentences in lines 193-196 as: "The root mean square error (RMSE) between the simulations and observations were adopted to evaluate the performance of Noah-MP. The average of the RMSEs of all the soil layers was defined as column RMSE (colRMSE)."

- The study of Li et al. (2015) is cited in line 200:

[revised manuscript text omitted]

The soil hydraulic parameters of each layer, including the porosity ($\theta_s$), saturated hydraulic conductivity ($K_s$), hydraulic potential ($\psi_s$), the Clapp-Hornberger parameter ($b$), field capacity ($\theta_{ref}$), wilt point ($\theta_w$), and saturated soil water diffusivity ($D_s$), were determined using the pedotransfer functions proposed by Hillel (1980), Cosby et al. (1984), and Wetzel and Chang (1987):

$$\theta_s = 0.489 - 0.00126(\%sand) \hspace{3cm} \text{(S1)}$$

$$K_s = 7.0556 \times 10^{-6.884 + 0.0153(\%sand)} \hspace{2.5cm} \text{(S2)}$$

$$\psi_s = -0.01 \times 10^{1.88 - 0.0131(\%sand)} \hspace{2.5cm} \text{(S3)}$$

$$b = 2.91 + 0.159(\%clay) \hspace{3cm} \text{(S4)}$$

$$\theta_{ref} = \theta_s \left[ \frac{1}{3} + \frac{2}{3} \left( \frac{5.79 \times 10^{-9}}{K_s} \right)^{1/(2b+3)} \right] \hspace{2cm} \text{(S5)}$$

$$\theta_w = 0.5\theta_s \left( \frac{-200}{\psi_s} \right)^{-1/b} \hspace{2.5cm} \text{(S6)}$$

$$D_s = b \cdot K_s \cdot \left( \frac{\psi_s}{\theta_s} \right) \hspace{3cm} \text{(S7)}$$

where $\%sand$ and $\%clay$ represent the percentage (%) of sand and clay content in soil, respectively.

**Table S1** Soil discretization scheme and soil particle fraction in this study.

| Layer | $Z_i$ | $\Delta Z_i$ | $Z_{h,i}$ | Sand (%) | Silt (%) | Clay (%) |
|---|---|---|---|---|---|---|
| 1 | 0.010 | 0.020 | 0.020 | 85.48 | 12.59 | 1.93 |
| 2 | 0.040 | 0.040 | 0.060 | | | |
| 3 | 0.090 | 0.060 | 0.120 | | | |
| 4 | 0.160 | 0.080 | 0.200 | 83.51 | 13.57 | 2.92 |
| 5 | 0.260 | 0.120 | 0.320 | 81.15 | 15.58 | 3.27 |
| 6 | 0.400 | 0.160 | 0.480 | 86.62 | 11.16 | 2.22 |
| 7 | 0.580 | 0.200 | 0.680 | 78.73 | 18.06 | 3.21 |
| 8 | 0.800 | 0.240 | 0.920 | 88.12 | 8.98 | 2.90 |
| 9 | 1.060 | 0.280 | 1.200 | 95.00 | 3.00 | 2.00 |
| 10 | 1.360 | 0.320 | 1.520 | | | |
| 11 | 1.700 | 0.360 | 1.880 | 92.50 | 4.00 | 3.50 |
| 12 | 2.080 | 0.400 | 2.280 | 90.00 | 5.00 | 5.00 |
| 13 | 2.500 | 0.440 | 2.720 | | | |
| 14 | 2.990 | 0.540 | 3.260 | | | |
| 15 | 3.580 | 0.640 | 3.900 | | | |
| 16 | 4.270 | 0.740 | 4.640 | 68.00 | 20.00 | 12.00 |
| 17 | 5.060 | 0.840 | 5.480 | | | |
| 18 | 5.950 | 0.940 | 6.420 | | | |
| 19 | 6.940 | 1.040 | 7.460 | | | |
| 20 | 7.980 | 1.040 | 8.500 | | | |

Layer node depth ($Z_i$), thickness ($\Delta Z_i$), and depth at layer interface ($Z_{h,i}$) for default soil column. All in meters.

[Figure]

**Figure. S1**. Monthly soil temperature (ST) at (a) 5 cm, (b) 25 cm, (c) 70 cm, (d) 140 cm, (e) 220 cm, (f) 300 cm at TGL site for observation (Obs), ensemble simulation considering snow (Sim-with snow), and ensemble simulation neglecting snow (Sim-no snow). The green and blue shadow represent the standard deviation of Sim-with snow and Sim-no snow experiments, respectively.

[Figure]

**Figure S2.** Distinction level for RMSE of ST at different layers during the warm season in the ensemble simulations. Limits of the boxes represent upper and lower quartiles, whiskers extend to the maximum and minimum RMSE. The black stations in the box are the average values. The lines in the box indicate the median value.

[Figure]

**Figure S3.** Distinction level for RMSE of ST at different layers during the cold season in the ensemble simulations. Limits of the boxes represent upper and lower quartiles, whiskers extend to the maximum and minimum RMSE. The black stations in the box are the average values. The lines in the box indicate the median value.

[Figure]

**Figure S4.** Distinction level for RMSE of SLW at different layers during the warm season in the ensemble simulations. Limits of the boxes represent upper and lower quartiles, whiskers extend to the maximum and minimum RMSE. The black stations in the box are the average values. The lines in the box indicate the median value.

[Figure]

**Figure S5.** Distinction level for RMSE of SLW at different layers during the cold season in the ensemble simulations. Limits of the boxes represent upper and lower quartiles, whiskers extend to the maximum and minimum RMSE. The black stations in the box are the average values. The lines in the box indicate the median value.

[Figure]

批注 [LX1]: deleted

**Figure.**  S6 Monthly  soil liquid water (SLW in %) at (a) 5 cm, (b) 25 cm, (c) 70 cm, (d) 140 cm, (e) 220 cm, (f) 300 cm for the RUN process.

[Figure]

批注 [LX2]: deleted

**Figure.**  S7 Monthly  soil liquid water (SLW in %) at (a) 5 cm, (b) 25 cm, (c) 70 cm, (d) 140 cm, (e) 220 cm, (f) 300 cm for the INF process.

[Figure]

批注 [LX3]: deleted

**Figure.**  S8 Monthly soil temperature at (a) 5 cm, (b) 25 cm, (c) 70 cm, (d) 140 cm, (e) 220 cm, (f) 300 cm for the TBOT process.

[Figure]

批注 [LX4]: deleted

**Figure.**  S9 Monthly soil temperature at (a) 5 cm, (b) 25 cm, (c) 70 cm, (d) 140 cm, (e) 220 cm, (f) 300 cm for the STC process.

---

## Author Response (AR2)

**Point-by-point response to referees**

We thank the reviewer for the insightful and constructive comments. We have made point-by-point responses and/or revisions according to your suggestions and instructions. We recall the comments of the reviewer in black, followed by our reply in blue.

**Responses to Referee #1**

**Report #1**

**Suggestions for revision or reasons for rejection (will be published if the paper is accepted for final publication)**
* * *
Thank you for your comments and effort you have put into reviewing the manuscript. The manuscript has been substantially revised following the two referee's suggestions. Please see List of changes at the end of this response.

**Responses to Referee #2**

**Report #2**

**Suggestions for revision or reasons for rejection (will be published if the paper is accepted for final publication)**

It's my pleasure to review gmd-2020-142 "Assessing the simulated soil thermal regime from Noah-MP LSM v1.1 for near-surface permafrost modeling on the Qinghai-Tibet Plateau" by Li et al. I don't think the authors have appropriately addressed my previous comments, and thus major revision is still recommended. I will echo my previous major comments below.

**Response:** Thanks very much for your time regarding our manuscript. We already conducted all the necessary analysis according to your comments.

For a more comprehensive assessment, we have added the two physical processes in the revised manuscript as suggested, i.e., the snow sublimation from wind (SUB) and

combination scheme by Li et al. (2020) (CMB). The general behaviors, influential processes, and sensitivities of the augmented Noah-MP for snow cover events, soil temperature and soil liquid water content have been tested and discussed in the revised manuscript.

Attachment is our detailed response to your concerns. With these revisions, we believe the quality of the manuscript has been greatly improved. We hope the reviewer can find that the comments have been addressed adequately, and we are happy to address additional concerns.

1. I note that there is a paper recently published by the same author to improve the performance of Noah-MP simulations on the same site. It will be interesting the authors firstly add their improvements, and then design more numerical experiments to test the uncertainties of different parameterization options. Since one additional site, soil moisture and snow measurements are available, the authors are suggested to also use these measurements to test the Noah-MP's performance. For the frozen soil, the soil moisture and soil temperature are fully coupled, which are also affected by the snow process, so it's also important to evaluate the performance of Noah-MP in simulating these variables.

I don't think "including augmentation work is out of scope of this paper". As shown in the paper published by the same author (Li, X., et al. (2020). Improving the Noah-MP model for simulating hydrothermal regime of the active layer in the permafrost regions of the Qinghai-Tibet Plateau. Journal of Geophysical Research: Atmospheres, 125), the cold bias noted for the Noah-MP (also found in this paper) is related to the underestimation of snow sublimation rate and inappropriate parameterizations of thermal roughness (z0h) and under-canopy aerodynamic resistance (ra,g). The authors showed in their previous publication that only introducing new parameterizations of snow sublimation, z0h and ra,g can improve the cold bias. Therefore, I don't found the necessary to test only the combination schemes of default Noah-MP if the new parameterizations are not included.

**Response:** Thank you for your comments.

We are sorry that we did not explain our previous work clearly. In our previous work, we only tested one selected combination of Noah-MP options (the options in bold in Table 2 in the study of Li et al. (2020)). Strictly, we didn't conclude that only by introducing new parameterizations of snow sublimation, $z_{0h}$ and $r_{a,g}$ can improve the cold bias since there are many other combinations are not assessed, which is one of our main purposes of the previous version of the manuscript.

We understand the referee's concerns. In the revised manuscript, we have added two physical processes as suggested for a more comprehensive assessment, i.e., the snow sublimation from wind (SUB) and combination scheme by Li et al. (2020) (CMB) (Table 1), in which users can turn on or off the snow sublimation from wind and the combination of thermal roughness ($z_{0h}$) and under-canopy aerodynamic resistance ($r_{a,g}$), respectively. Our main conclusion is that the SUB process together with the snow/soil temperature time scheme (STC) play a dominant role for snow simulation. The combination of $z_{0h}$ and $r_{a,g}$ helps to elevate soil temperature. Details can be seen in Section 3.2.1, 3.2.2 and 3.3.

With these revisions, we believe the reviewer and potential readers can understand the differences between the present and our previous studies, and the novelty of this study is more clear.

In addition, I don't agree with the authors that "only focus on one site is enough to provide a reference for simulating permafrost state on the Tibetan Plateau". As also noted by the authors, there may be different environmental controlling factors among the sites, so including additional sites will make the conclusions more useful and general for the permafrost simulation on the Tibetan Plateau.

**Response:** We agree that add more sites would strengthen our conclusions.

We also tested Noah-MP model at the BLH station. Our main findings at BLH site are basically consistent with that at TGL site (see below). However, we realized that this will make our manuscript very long. Our main goal is 1) to investigate the general performance and sensitivity of Noah-MP model with all possible combinations for soil hydrothermal simulations, and 2) to present a reference to better understand the land surface processes in the permafrost regions of the QTP. We tried our best to make this manuscript concise and we are afraid that add one more site would be distractive to potential readers from our main goals. Therefore, we would rather focus on one site in this manuscript.

The results and conclusions at BLH sites are attached in the supplementary file as follows:

Our main findings at BLH site are:

(1) Noah-MP tend to overestimate snow cover events at BLH site with large uncertainties during the cold months (Nov.-Mar.). Moreover, snow cover events are mostly influenced by the STC and SUB process (Figure 3), and the combination of STC(1) and SUB(2) tend to produce better results (Figure 8). The small influence of physical processes during the warm season (Figure 3c) is because there are limited snow events, and its inability of reproducing snow cover in May (Figure 1).

(2) Noah-MP generally underestimate STs with relatively large gaps during the snow-affected months (Nov.-Mar.), and the simulated ST in the snow-affected months (Nov.-Mar.) showed relatively wide uncertainty ranges (Figure 2). STs is mostly influenced by the snow processes, i.e. the STC and SUB process (Figure 4), especially during the cold season. In the warm season, the SFC and RUN process dominate the simulation of STs (Figure 4c). The combination of roughness length for heat and under-canopy aerodynamic resistance contributes to elevated soil temperature (Figure 9).

(3) Noah-MP totally underestimate SLW at BLH site (Figure 2). The RUN process dominates the SLW at most layers simulation with limit impacts.

- **General performance of the ensemble simulation**

[Figure]

**Figure 1.** Monthly variations of ground albedo at BLH site for observation (Obs), and the ensemble simulation (Sim). The light blue shadow represents the standard deviation of the ensemble simulation.

[Figure]

**Figure 2.** Monthly soil temperature (ST in °C) and soil liquid water (SLW in %) at (a, g) 5 cm, (b, h) 25 cm, (c, i) 70 cm, (d, j) 140 cm, (e, k) 220 cm, (f, l) 300 cm at BLH site. The light blue shadow represents the standard deviation of the ensemble simulation. The black line-symbol represents the ensemble mean of simulations with STC(1) and SUB(2).

- **Influence degrees of physical processes**

[Figure]

**Figure 3.** The maximum difference of the mean overall accuracy (OA) for albedo (ALB-$\Delta OA$) in each physical process during the (a) annual, (b) cold season, and (c) warm season at BLH site.

[Figure]

**Figure 4.** The maximum difference of the mean RMSE for (a, c and e) soil temperature (ST-$\Delta \overline{RMSE}$ in °C) and (b, d and f) soil liquid water (SLW-$\Delta \overline{RMSE}$ in %) in each physical process during the (a and b) annual, (c and d) warm, and (e and f) cold season at different soil depths at BLH site.

- **Sensitivities of physical processes and general behaviors of parameterizations**

[Figure]

**Figure 5.** Distinction level for overall accuracy (OA) of snow cover events (SCEs) during the annual, warm, and cold seasons in the ensemble simulations at BLH site.

Limits of the boxes represent upper and lower quartiles, lines in the box indicate the median value.

[Figure]

**Figure 6.** Distinction level for RMSE of ST at different layers during the annual, warm, and cold seasons in the ensemble simulations at BLH site. Limits of the boxes represent upper and lower quartiles, lines in the box indicate the median value.

[Figure]

**Figure 7.** Distinction level for RMSE of SH2O at different layers during the annual, warm, and cold seasons in the ensemble simulations at BLH site. Limits of the boxes represent upper and lower quartiles, lines in the box indicate the median value.

[Figure]

**Figure 8.** Uncertainty interval of ground albedo at BLH site in dominant physical processes (STC and SUB) for snow cover event simulation.

[Figure]

**Figure 9.** Monthly soil temperature (ST in °C) at (a) 5 cm, (b) 25 cm, (c) 70 cm, (d)

140 cm, (e) 220 cm, (f) 300 cm for the SFC process that consider the CMB(2) and STC(1)+SUB(2) processes or not.

Also, since the soil moisture and soil temperature are fully coupled in the permafrost areas, I think both soil moisture and soil temperature should be discussed in detail, and thus the title can be also revised accordingly.

**Response:** The general behaviors, influential processes, and sensitivities of Noah-MP for soil temperature and soil moisture (represented by soil liquid water content since soil ice could not be recorded using the observation equipment) are tested and discussed in the revised manuscript. Accordingly, the title has been revised as "Assessing the simulated soil hydrothermal regime of active layer from Noah-MP LSM v1.1 in the permafrost regions of the Qinghai-Tibet Plateau".

2. As shown by the authors, "Noah-MP greatly overestimates snow cover both in magnitude and duration, inducing huge cold bias and large uncertainties in soil temperatures", and these results are contrary to the reality. I don't think this is the reason to ignore the snow process, since snow is widely presented in the permafrost areas. Instead, the authors need to include their new parameterization of snow sublimation in previous publication, and then test its performance together with the ALB and SNF options.

**Response:** Thank you for your comments. We considered the snow sublimation, ALB and SNF options in the revised version. We found that snow cover events are mostly affected by the snow sublimation process (SUB) and the snow/soil temperature time scheme (STC). The influence of ALB and SNF on snow cover events is significant but limited. Moreover, the performance orders followed SUB(2) > SUB(1), STC(1) > STC(2), ALB(2) > ALB (1), SNF(3) > SNF(2) >SNF(1).

The manuscript has been greatly changed, we would not copy the text below. Please refer to Table 1, Section 3.2.1, 3.2.2 and 3.3 for details.

3. It's still strange to me that the impact of RUN is so important for the soil temperature simulations at both cold and warm seasons. So, it will be more useful the authors investigate both the soil moisture and temperature simulations in detail, then the authors may provide appropriate explanation on this.

**Response:** One thing should be noted that we use soil liquid water (SLW) as an alternative to investigate the soil moisture (SLW + ice) since soil ice could not be recorded using the observation equipment.

Soil moisture refers to the total water in the soil. In the warm season, soil moisture is equal to the SLW. In the cold season, the soil moisture (SLW + ice) was nearly identical to SLW at the end of the warm season.

Our results showed that the four schemes of RUN process performed differently for SLW simulation in the warm season (Figure. S1) and thus soil moisture (SLW + ice) in the cold season.

Different SLWs in the warm season result in the difference of the surface energy partitioning and thus different soil temperatures. Generally, higher estimation of SLWs induce greater sensible heat and thus smaller soil temperature (Gao et al., 2015). In the cold season, much of the liquid water freezes into ice, which would greatly influence the thermal conductivity of frozen soil considering thermal conductivity of ice is nearly four times that of the equivalent liquid water. Therefore, the impact of RUN is important for the soil temperature simulations at both warm and cold seasons.

To be clear, we added relevant explanations in section 4.2.2 of the revised version.

[Figure]

**Figure. S1** Monthly soil liquid water (SLW in %) at (a) 5 cm, (b) 25 cm, (c) 70 cm, (d) 140 cm, (e) 220 cm, (f) 300 cm for the RUN process.

**References:**

Gao, Y., Kai, L., Fei, C., Jiang, Y., and Lu, C.: Assessing and improving Noah-MP land model simulations for the central Tibetan Plateau, J. Geophys. Res.-Atmos., 120, 9258-9278, https://doi.org/10.1002/2015JD023404, 2015.

4. Detailed information is needed on how the authors revise the soil moisture and heat flow equations when the simulation depth was extended to 8.0 m and soil column was discretized into 20 layers. How the authors define the bottom boundary for the soil moisture and heat flow simulations?

**Response:** The equations for soil moisture and temperature are not modified and followed the default Richards' equation and 1-d heat conduction equation, respectively. The lower boundary conditions follow the default settings of Noah-MP.

For the heat flow simulation, the bottom boundary condition depend on the scheme in TBOT process: (1) zero heat flux; or (2) soil temperature at 8 m depth (usually using the annual-mean 2-m air temperature) (Niu and Yang, 2011).

For the soil moisture simulation, the recharge of groundwater is not considered because

of the existence of permafrost. The bottom boundary condition depend on the scheme in RUN process: (1) SIMGM: TOPMODEL-based runoff with the simple groundwater (Niu et al., 2007); (2) SIMTOP: TOPMODEL-based runoff with an equilibrium water table (Niu et al., 2005); (3) Schaake96: Infiltration-excess-based surface runoff with free drainage (Schaake et al., 1996); (4) BATS: BATS runoff with free drainage (Yang & Dickinson, 1996).

What we have modified to the model itself is setting the corresponding soil parameters for each layer instead of using the same values. Technically, we changed the soil parameter variables from REAL types into REAL ARRAY types, and calculate soil hydrothermal parameters of each layer using a loop structure.

**References:**

Niu, G., & Yang, Z. (2011). The community Noah land-surface model (LSM) with multi-physics options: User's guide. http://www.jsg.utexas.edu/noah-mp/files/Users_Guide_v0.pdf

Niu, G.-Y., Yang, Z.-L., Dickinson, R. E., Gulden, L. E., & Su, H. (2007). Development of a simple groundwater model for use in climate models and evaluation with Gravity Recovery and Climate Experiment data. Journal of Geophysical Research, 112, D07103. https://doi.org/10.1029/2006JD007522

Niu, G.-Y., Yang, Z.-L., Dickinson, R. E., & Gulden, L. E. (2005). A simple TOPMODEL-based runoff parameterization (SIMTOP) for use in global climate models. Journal of Geophysical Research, 110, D21106. https://doi.org/10.1029/2005JD006111

Schaake, J. C., Koren, V. I., Duan, Q. Y., Mitchell, K., & Chen, F. (1996). Simple water balance model for estimating runoff at different spatial and temporal scales. Journal of Geophysical Research, 101(D3), 7461–7475. https://doi.org/10.1029/95JD02892

Yang, Z.-L., & Dickinson, R. E. (1996). Description of the Biosphere-Atmosphere Transfer Scheme (BATS) for the soil moisture workshop and evaluation of its performance. Global and Planetary Change, 13(1-4), 117–134. https://doi.org/10.1016/0921-8181(95)00041-0

**List of changes**

1. Revised title as " Assessing the simulated soil hydrothermal regime of active layer from Noah-MP LSM v1.1 in the permafrost regions of the Qinghai-Tibet Plateau"

2. Two physical processes, i.e., the snow sublimation from wind (SUB) and combination scheme by Li et al. (2020) (CMB) are included to obtain a more comprehensive assessment. The general behaviors, influential processes, and sensitivities of the augmented Noah-MP for snow cover events, soil temperature and soil liquid water content during warm and cold seasons are tested and discussed in the revised manuscript.

3. The purpose of this study is assess the model structure of Noah-MP without considering the uncertainties of forcing data and model parameters. Only VEG(1) is adopted in the VEG process.

4. Deleted the optimal combination part.

5. Newly added section 3.3, in which the influence of snow cover and surface drag coefficient on soil hydrothermal dynamics are analyzed.

6. Discussed the snow cover on the QTP and its influence on soil hydrothermal regime in section 4.1

7. All typos have been corrected.

8. All "soil thermal" in the manuscript has been revised as "soil hydrothermal"

---

## Author Response (AR3)

**Point-by-point response to referees**

We thank the reviewer for the insightful and constructive comments. We have made point-by-point responses and/or revisions according to your suggestions and instructions. We recall the comments of the reviewer in black, followed by our reply in blue.

**Responses to Referee #2**

**Report #2**

**Suggestions for revision or reasons for rejection (will be published if the paper is accepted for final publication)**

It's my pleasure to review gmd-2020-142 "Assessing the simulated soil hydrothermal regime of active layer from Noah-MP LSM v1.1 in the permafrost regions of the Qinghai-Tibet Plateau" by Li et al. The authors have appropriately addressed my previous comments, and the paper can be accepted after addressing my following two minor concerns.

1. Since the authors have also tested the performance of Noah-MP in the other site, i.e., BLH site, it's suggested to briefly describe related conclusions in the Discussion part to highlight the transferability of key findings to the other site. The results related to the test in BLH site can be included in the Supplementary materials as well.

**Response:** Thank you for your comment. We already added the main findings at BLH site in the Supplementary materials. Also, we added some explanations in the Perspective part as follows:

"Further tests at another permafrost site (BLH site, 34.82°N, 92.92°E, Alt.: 4,659 m a.s.l) basically showed consistent conclusions with that at TGL site (see Supplementary files for details), indicating that relevant results and methodologies can be practical guidelines for improving the parameterizations of physical processes and testing their uncertainties towards soil hydrothermal modeling in the permafrost regions of the plateau."

2. I note that the liquid water in the top two soil layers are generally underestimated especially at the BLH site, is this related to the ignore of soil organic matter effect as evidenced by many other researchers such as Yang et al. (2009), Chen et al. (2012), and Zheng et al. (2015)?

Yang, K., Koike, T., Ye, B., and Bastidas, L.: Inverse analysis of the role of soil vertical heterogeneity in controlling surface soil state and energy partition, J. Geophys. Res.-Atmos., 110, D08101, 2005.

Chen, Y., Yang, K., Tang, W., Qin, J., and Zhao, L.: Parameterizing soil organic carbon's impacts on soil porosity and thermal parameters for Eastern Tibet grasslands, Sci. Chin. Earth Sci., 55, 1001-1011, 2012.

Zheng, D., van der Velde, R., Su, Z., Wang, X., Wen, J., Booij, M. J., Hoekstra, A. Y., and Chen, Y.: Augmentations to the Noah model physics for application to the Yellow River source area. Part I: Soil water flow, Journal of Hydrometeorology, 16(6), 2659-2676, 2015.

**Response:** Thank you for your insightful comment. Underestimation of topsoil moisture on the QTP is a common problem in many LSMs, which can be attributed to the poor representation of the complex soil profiles in current models (Yang et al., 2005). The missing of soil organic matter can be one of the reasons. However, some studies also illustrated the limited impacts of soil organic matter on soil moisture (Gao et al., 2015; Sun et al., 2017; Li et al., 2020) in both the plateau and other regions (Khlosi et al., 2013; Rawls et al., 2003). Moreover, many other studies highlighted the mucilage in the rhizosphere (Gao et al., 2015), the gravels (Yi et al., 2018), and the permafrost (Wu et al., 2018) in soil moisture simulation. It is worth quantifying the influence of these physics on the simulation of soil moisture in the future.

**References:**

Gao, Y., Kai, L., Fei, C., Jiang, Y., and Lu, C.: Assessing and improving Noah-MP land model simulations for the central Tibetan Plateau, J. Geophys. Res.-Atmos., 120, 9258-9278, https://doi.org/10.1002/2015JD023404, 2015.

Khlosi, M., Cornelis, W., Douaik, A., Hazzouri, A., Habib, H., & Gabriels, D. (2013).

Exploration of the interaction between hydraulic and physicochemical properties of Syrian soils. Vadose Zone Journal, 12(4), vzj2012.0209. https://doi.org/10.2136/vzj2012.0209

Li, X., Wu, T., Zhu, X., Jiang, Y., Hu, G., Hao, J., Ni, J., Li, R., Qiao, Y., Yang, C., Ma, W., Wen, A., and Ying, X.: Improving the Noah-MP Model for simulating hydrothermal regime of the active layer in the permafrost regions of the Qinghai-Tibet Plateau, J. Geophys. Res.-Atmos., 125, e2020JD032588, https://doi.org/10.1029/2020JD032588, 2020.

Rawls, W. J., Pachepsky, Y. A., Ritchie, J. C., Sobecki, T. M., & Bloodworth, H. (2003). Effect of soil organic carbon on soil water retention. Geoderma, 116(1–2), 61–76. https://doi.org/10.1016/S0016-7061(03)00094-6

Sun, Y., Wang, Y., Sun, Z., Liu, G., & Gao, Z. (2017). Impact of soil organic matter on water hold capacity in permafrost active layer in the Tibetan Plateau. Journal of Desert Research, 37(2), 288–295. https://doi.org/10.7522/j.issn.1000-694X.2016.00083

Wu, X. B., Nan, Z. T., Zhao, S. P., Zhao, L., and Cheng, G. D.: Spatial modeling of permafrost distribution and properties on the Qinghai-Tibet Plateau, Permafr. Periglac. Process., 29, 86-99, https://doi.org/10.1002/ppp.1971, 2018.

Yang, K., Koike, T., Ye, B., and Bastidas, L.: Inverse analysis of the role of soil vertical heterogeneity in controlling surface soil state and energy partition, J. Geophys. Res.-Atmos., 110, D08101, 2005.

Yi, S., He, Y., Guo, X., Chen, J., Wu, Q., Qin, Y., & Ding, Y. (2018). The physical properties of coarse-fragment soils and their effects on permafrost dynamics: A case study on the central Qinghai-Tibetan Plateau. The Cryosphere, 12(9), 3067– 3083. https://doi.org/10.5194/tc-12-3067-2018

**List of changes**

1. Brief descriptions about the conclusions at BLH site in the Perspective part and detailed results in the Supplementary file.

2. Two co-authors were added for their contributions to this work.

3. Thanks to the supports of the National Natural Science Foundation of China (42071093), the CAS "Hundred Talents" Program (Sizhong Yang), and the National Cryosphere Desert Data Center Program (E0510104).

---

## Author Response (AR4)

**Point-by-point response to editor**

We thank the editor for the comments. We have made point-by-point responses and/or revisions according to your suggestions and instructions. We recall the comments of the editor in black, followed by our reply in blue. Please note that the line numbers in our responses refer to the new line numbers.

Comments to the Author:

Dear authors,

Many thanks for your quick reply. Below my comments on your submitted version:

- Line 46: Such changes have...

**Response:** Corrected as suggested.

- Line 134, 179, 180, labels in the Y-axis in Figure S10, etc.: a blank space between numerals and units is mandatory (e.g., 25 cm)

**Response:** Corrected as suggested.

Spaces between number and unit are also added in line 124, 126, 218, 243, 244, 245, 290, 294, 348, 349, 356, 357, 383, 406, 407, and labels in the Y-axis in Figure 3.

Besides, a space between degree sign and direction of coordinates are added in line 118 and 571.

- According to our journal guidelines, it is necessary that you publish the code of your modified version of Noah-MP with the paper. You can do it as supplementary material to your paper or publishing it in a Zenodo repository.

**Response:** Thanks very much for your professional comment. We have uploaded the modified Noah-MP in the Zenodo repository (http://doi.org/10.5281/zenodo.4555449). And the Code availability and Data availability section has revised as following:

"*Code availability*. The original source code of the offline 1D Noah-MP LSM v1.1 is available at

https://ral.ucar.edu/solutions/products/noah-multiparameterization-land-surface-model-noah-mp-lsm (last access: 23 February 2021). The modified Noah-MP with the consideration of vertical heterogeneity in soil profile, snow sublimation from wind and the combination of roughness length for heat and under-canopy aerodynamic resistance can be downloaded at http://doi.org/10.5281/zenodo.4555449."

"*Data availability*. The 1-hourly forcing data, daily soil temperature and liquid water content data at the TGL and BLH sites are available at https://doi.org/10.17632/h7hbd69nnr.2. Soil texture data can be obtained at https://doi.org/10.1016/j.catena.2017.04.011 (Hu et al., 2017). The AVHRR LAI data can be downloaded from https://www.ncei.noaa.gov/data/ (Claverie et al., 2016)."

- Line 620: It is GNU/Linux, not Linux. You have used the full system, not just the kernel. And please clarify what kind of distribution and version you have used (Debian, RedHat, CentOS...)

**Response:** Thanks very much for your professional comment. We have revised the sentence as "JC helped to compile the model in a GNU/Linux (CentOS 7.0) environment." in line 621.

It is extraordinary to add authors to the manuscript at this stage, after an almost completed review process. Since the last version, you have not conducted new experiments, and therefore no additional compilation of the model or additional simulations seem to have been necessary. I have to say that I need a better explanation of what has happened and why you forgot to add them to the previous versions. And I want to make clear that although there is nothing that a priori can prevent, this does not grant that we accept such change at this stage without a good reason.

**Response:** Thanks very much for your comments. We apologize for the author changes.

In fact, Dr. Sizhong Yang and Mr. Guohui Zhao have been involved in this work since the Major Revision round. In a webinar, Dr. Sizhong Yang, a recent colleague of our group, suggested to illustrate the results and discussions of the two newly-added physical processes in separate paragraphs, i.e., Section 3.3 and Section 4.1. He also suggested to remove the optimal combination part to sharpen the focus of the manuscript. His suggestions directly responded to the reviewer's concerns and made the structure of the manuscript clearer. Mr. Guohui Zhao, engineer of the data center, provided convenience in the application, allocation and management of computation resources in the supercomputing system, which enabled our subsequent deployment and simulation work to be carried out as soon as possible.

We believe they made intellectual contribution to this work. However, we did not include Dr. Sizhong Yang because he was moving to a new institute, and there were some problems in the human resource department. Therefore, we did not know how to do with this. Until recently, Dr. Sizhong Yang solved this problem, and thus we hope we could include him. When compiling and running the model, we directly approached Dr. Jie Chen. He got a lot of support in terms of applying for computing nodes from Mr. Guohui Zhao. We would like to reflect his contribution in this work.

However, we agree with you that we should not do this at this stage. Therefore, we would like to thank their contributions in the Acknowledgment. Sorry for the inconvenience.

**List of changes**

1. Uploaded the modified Noah-MP in Zenodo.

2. Removal of Dr. Sizhong Yang and Mr. Guohui Zhao in the co-authors list, and thank them in the Acknowledgement section.

3. Added blank spaces between numbers and units.

4. Deleted reference that is not cited in the text, and cheked the format of References and In-text citations.

5. Corrected all typos.